# A Survey on Large Language Model-Based Social Agents in Game-Theoretic Scenarios

**Xiachong Feng**[μ]   **Longxu Dou**[s]   **Ella Li**[αγ]   **Qinghao Wang**[δ]   **Haochuan Wang**[β]
**Yu Guo**[β]   **Chang Ma**[μ]   **Lingpeng Kong**[μ]
[μ]The University of Hong Kong   [s]Independent Researcher   [α]National University of Singapore
[γ]Institute for Infocomm Research (I2R), A*STAR   [δ]Peking University   [β]Harbin Institute of Technology
fengxc@hku.hk,lpk@cs.hku.hk

Reviewed on OpenReview: https://openreview.net/forum?id=CsoSWpR5xC

## Abstract

Game-theoretic scenarios have become pivotal in evaluating the social intelligence of Large Language Model (LLM)-based social agents. While numerous studies have explored these agents in such settings, there is a lack of a comprehensive survey summarizing the current progress. To address this gap, we systematically review existing research on LLM-based social agents within game-theoretic scenarios. Our survey organizes the findings into three core components: Game Framework, Social Agent, and Evaluation Protocol. The game framework encompasses diverse game scenarios, ranging from choice-focusing to communication-focusing games. The social agent part explores agents' preferences, beliefs, and reasoning abilities, as well as their interactions and synergistic effects on decision-making. The evaluation protocol covers both game-agnostic and game-specific metrics for assessing agent performance. Additionally, we analyze the performance of current social agents across various game scenarios. By reflecting on the current research and identifying future research directions, this survey provides insights to advance the development and evaluation of social agents in game-theoretic scenarios.

## 1 Introduction

The rapid advancement of Large Language Models (LLMs) (Achiam et al., 2023; Team et al., 2023; Jiang et al., 2023; Yang et al., 2024a; Dubey et al., 2024) has achieved exceptional performance across a wide array of applications, including personal assistant (Li et al., 2024b), search engines (Chen et al., 2024b), code generation (Wang et al., 2024b) and embodied intelligence (Liu et al., 2024a). Building on this capability, a growing area of research focuses on employing LLMs as central controllers to develop autonomous agents with human-like decision-making abilities (Sumers et al., 2023; Wang et al., 2024a). This progress brings the realization of Artificial General Intelligence (AGI) within reach (Bubeck et al., 2023), paving the way for a future where human-AI interaction, collaboration, and coexistence shape a shared, symbiotic society (Mahmud et al., 2023; Ren et al., 2024). Therefore, it is crucial to evaluate and enhance the *social intelligence* of AI, particularly LLM-based social agents, as it determines their ability to engage effectively in sophisticated social scenarios (Mathur et al., 2024).

Social intelligence is the foundation of all successful interpersonal relationships and is also a prerequisite for AGI (Hunt, 1928; Kihlstrom & Cantor, 2000; Hovy & Yang, 2021). Drawing on insights from both social science and AI research, Li et al. (2024a) has established a comprehensive Social AI Taxonomy, which categorizes social intelligence into three dimensions: *situational intelligence*, the ability to comprehend the social environment (Derks et al., 2007); *cognitive intelligence*, the ability to understand others' intents and beliefs (Barnes & Sternberg, 1989); and *behavioural intelligence*, the ability to behave and interact appropriately (Ford & Tisak, 1983). To evaluate artificial social intelligence, researchers have conducted extensive studies, with particular focus on *game-theoretic scenarios*, as these studies simultaneously encompass all

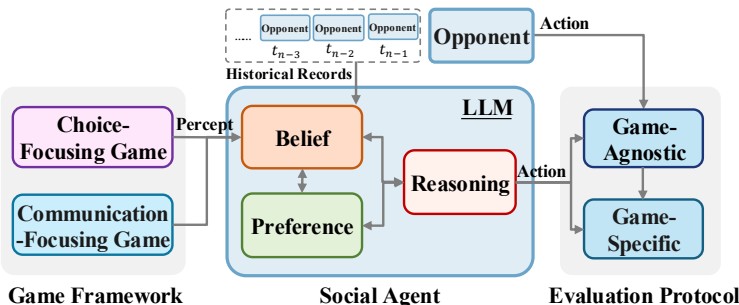

Figure 1: Taxonomy of LLM-based social agents in game-theoretic scenarios.

above three dimensions of social intelligence (Aher et al., 2022; Horton, 2023; Phelps & Russell, 2023; Akata et al., 2023; Brookins & DeBacker, 2023).

Game theory, a long-established field in microeconomics, offers a robust mathematical framework for analyzing social interactions among cooperating and competing players, with wide-ranging applications (Fudenberg & Tirole, 1991; Camerer, 2011). Specifically, evaluations in game-theoretic scenarios require social agents to understand the game scenario, infer opponents' actions, and adopt appropriate responses, representing an advanced form of social intelligence (Van Der Hoek et al., 2005; Zhang et al., 2024b). Moreover, the multi-agent participation and dynamic nature of the environment in game scenarios present additional challenges for social agents. Consequently, extensive research has examined social agents within game-theoretic scenarios, offering substantial empirical evidence for understanding their social intelligence (Guo, 2023; Meng, 2024; Mei et al., 2024). However, there is currently a lack of a comprehensive review that summarizes the current progress in this area and considers future directions.

To address this gap, we have thoroughly reviewed the existing research on LLM-based social agents in game-theoretic scenarios and have organized the findings according to a meticulously designed taxonomy, as illustrated in Figure 1. Specifically, the taxonomy comprises three main components: Game Framework (§2), Social Agent (§3), and Evaluation Protocol (§4). The Game Framework section includes two parts: Choice-Focusing Game (§2.1) and Communication-Focusing Game (§2.2). *Choice-Focusing Game* refers to a series of scenarios where participants engage with little to no communication, such as prisoner's dilemma (Brookins & DeBacker, 2023) and poker (Yim et al., 2024). *Communication-Focusing Game* refers to games where communication among participants is a core component, such as negotiation (Bianchi et al., 2024) and diplomacy (Bakhtin et al., 2022). The Social Agent section comprises four parts: Preference Module (§3.1), Belief Module (§3.2), Reasoning Module (§3.3), and PBR-Triangular Interaction (§3.4). *Preference Module* focuses on research analyzing the intrinsic preferences of LLMs and their ability to follow internal or pre-defined preferences (Guo, 2023). *Belief Module* explores studies on the internal beliefs of models, belief enhancement, and belief revision (Fan et al., 2023). *Reasoning Module* examines research on strategic reasoning, particularly involving theory-of-mind capabilities and reinforcement learning (Guo et al., 2023). *PBR-Triangular Interaction* focus on the interaction among different modules and their influence on final decision-making. The Evaluation Protocol section comprises three components: Game-Agnostic Evaluation (§4.1), Game-Specific Evaluation (§4.2), and Performance Assessment of Social Agents (§4.3). *Game-Agnostic Evaluation* focuses on universal metrics that can be used to assess game outcomes (Duan et al., 2024b). *Game-Specific Evaluation* emphasizes context-specific metrics tailored to the evaluation dimensions of particular game scenarios (Qi et al., 2024). *Performance Assessment of Social Agents* summarizes the performance of current social agents across various game scenarios and analyzes the strengths and weaknesses of these agents, as well as their comparison with human players.

Based on the above taxonomy, we provide a detailed summary of current research progress, reflect on each part, and offer insights into potential future research directions (§6), with the aim of inspiring further studies in this evolving field.

We summarize the core contributions of this survey as follows: (1) *A well-structured literature taxonomy*: We conduct a comprehensive review and categorization of existing research on social agents in game-theoretic

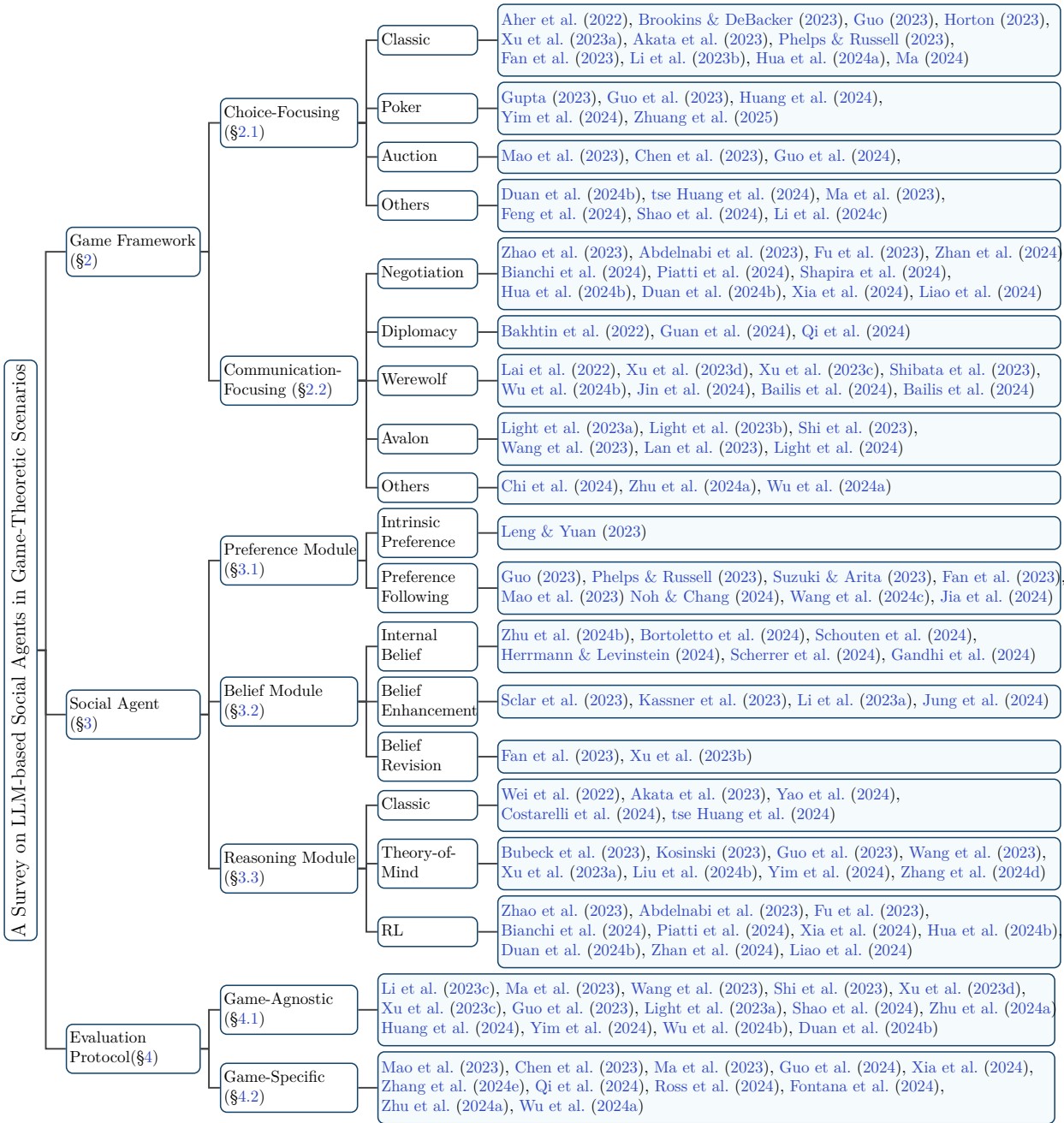

Figure 2: Taxonomy of recent research on LLM-based social agents in game-theoretic scenarios.

settings, providing a clear framework to support future research positioning. (2) *A unified and comprehensive performance comparison*: We summarize the performance of current social agents across a range of games, identifying both strengths and limitations in different scenarios to guide subsequent investigations. (3) *Detailed development guidelines*: Drawing on existing findings, we offer practical research recommendations from both the design and evaluation perspectives. (4) *Concrete future directions*: We highlight current research gaps and propose feasible future directions along with preliminary solutions to encourage continued exploration in this area.

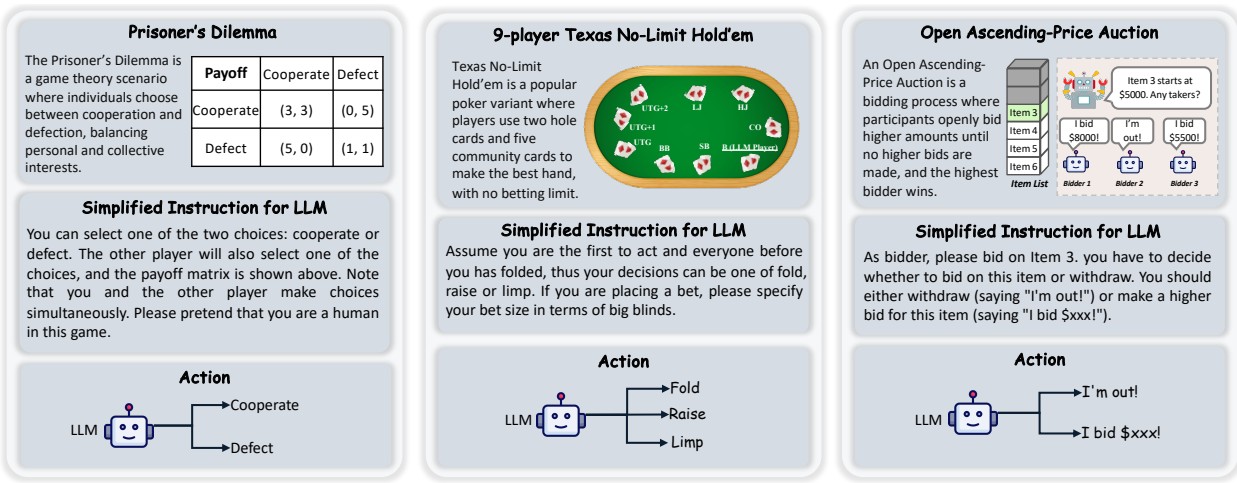

Figure 3: Illustration of choice-focusing games.

## 2 Game Framework

In this section, we describe the game-theoretic scenarios explored in existing research, including both choice-focusing games and communication-focusing games.

### 2.1 Choice-Focusing Game

Choice-focusing games are game-theoretic scenarios in which participants make decisions based primarily on observable actions and environmental conditions, with minimal or no communication involved. Existing research focuses on social agents in three types of choice-focusing scenarios: *classic game-theoretic games*, *poker*, and *auctions*. Some game examples are shown in Figure 3. Figure 4 presents simple definitions of different types of games.

Classic game-theoretic games, such as the prisoner's dilemma, have been distilled by economists from various real-world situations. These games are well-defined, with rigorous mathematical foundations, and can be extended to numerous scenarios (Owen, 2013). Consequently, many studies have utilized these games as testbeds to study social agents. The prisoner's dilemma (Rapoport & Chammah, 1965), as the most famous and widely recognized game, has been extensively utilized in numerous studies. Brookins & DeBacker (2023) and Guo (2023) evaluated the strategic reasoning capabilities of GPT-3.5 and GPT-4, respectively, in the classic prisoner's dilemma, highlighting the sensitivity of LLM responses to input instructions, which contributes to low output robustness. This underscores the critical need for future evaluations to focus on instruction robustness testing. Furthermore, Akata et al. (2023) and Phelps & Russell (2023) extended their analyses to the iterated prisoner's dilemma, investigating the ability of LLMs to optimize decision-making by utilizing historical information. Interestingly, Brookins & DeBacker (2023) observed that GPT-3.5 replicates human tendencies toward fairness and cooperation, whereas Akata et al. (2023) found GPT-4 to be less tolerant and more rigid in its decision-making. Additionally, Xu et al. (2023a) studied a more complex multi-player iterative prisoner's dilemma scenario within a multi-agent framework driven by LLMs. In addition to the prisoner's dilemma, numerous studies have also employed various classic game-theoretic games as foundational frameworks for research, including the Dictator Game (Horton, 2023; Fan et al., 2023; Brookins & DeBacker, 2023; Ma, 2024), Ultimatum Game (Aher et al., 2022; Guo, 2023), Public Goods Game (Li et al., 2023b; Xu et al., 2023a), Battle of the Sexes (Akata et al., 2023), Rock-Paper-Scissors (Fan et al., 2023), and Ring-Network Games (Fan et al., 2023).

Poker is a globally popular card game with numerous variations (Waterman, 1970). Winning in poker often requires astute strategic reasoning, as it is a non-cooperative, imperfect information, and dynamic game (Moravčík et al., 2017; Huang et al., 2024). Consequently, many researchers evaluate social agents by

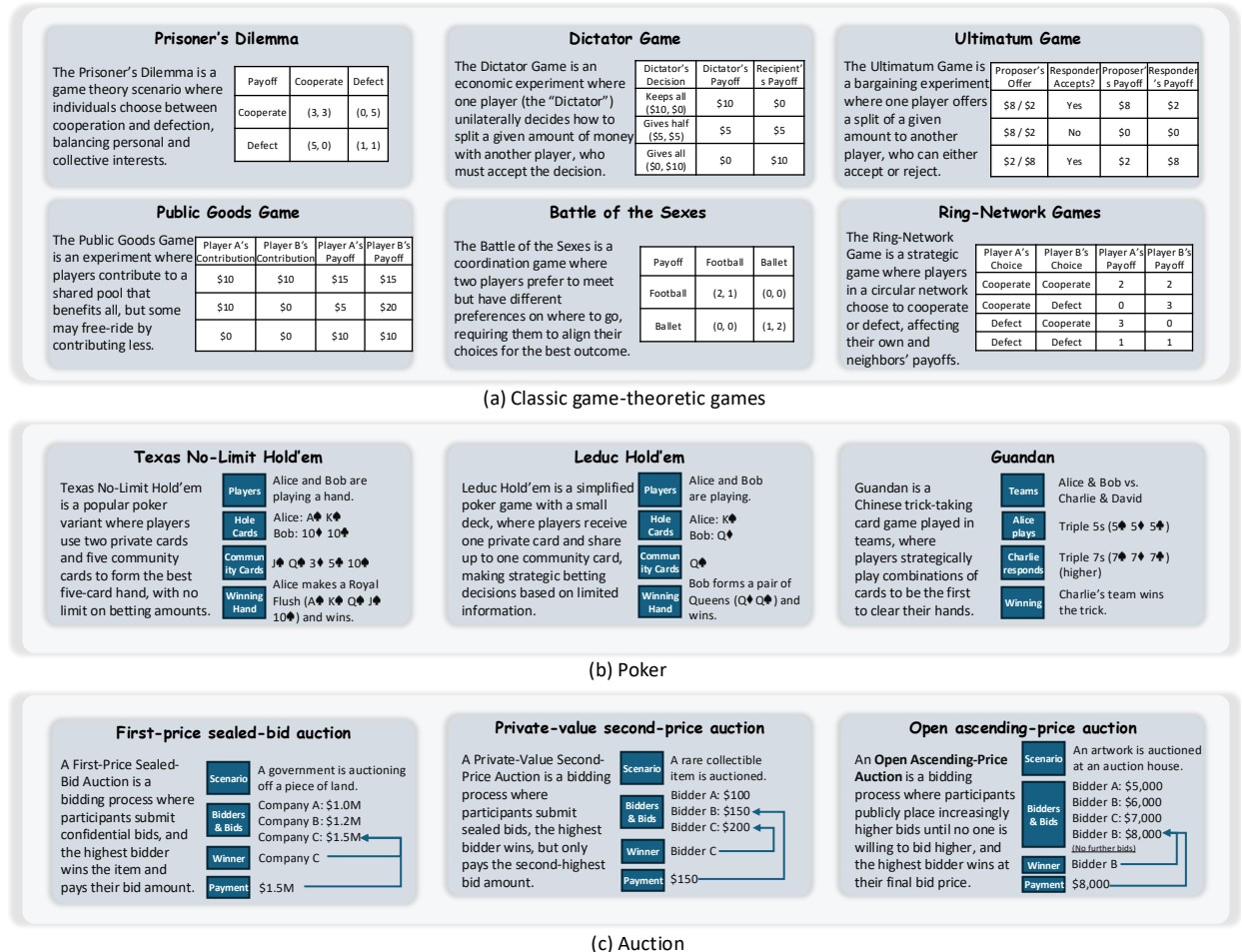

Figure 4: Introduction to different types of game theory games.

assessing their performance as poker players. Gupta (2023) studied 9-player Texas No-Limit Hold'em and concluded that the performance of both ChatGPT and GPT-4 is not game-theory optimal. Furthermore, their findings highlight the divergent poker tactics of the two models: ChatGPT's conservativeness contrasts sharply with GPT-4's aggression. Guo et al. (2023) conducted research on Leduc Hold'em, developing a social agent, Suspicion-Agent, which outperformed traditional reinforcement learning-based agents in poker. They also noted two critical issues: the outputs of LLMs are highly sensitive to the prompts, and the quality of the model's output declines rapidly as the prompt length increases. Yim et al. (2024) focused on Guandan, currently the most popular poker game in China, to investigate cooperative strategies in poker within a Chinese-language context. Interestingly, their experimental results show that while LLMs currently fall short of reinforcement learning models in performance, they underscore the future potential of LLMs in this domain. To provide a more comprehensive evaluation of the poker-playing abilities of LLMs, Zhuang et al. (2025) introduced PokerBench, a benchmark comprising 11,000 decision-making scenarios in poker, covering an exhaustive range of game situations, including 1,000 pre-flop and 10,000 post-flop scenarios. Poker is a complex game, and investigating whether social agents exhibit behavioural patterns that enable foresighted cooperation and competition in poker presents an intriguing avenue for future research.

Auction is a competitive process in which participants place bids on an item, providing a rich environment for evaluating strategic planning, resource allocation, risk management, and competitive behaviours (Kagel & Levin, 1986). As a typical non-cooperative game with incomplete information, it has garnered significant attention from researchers. Mao et al. (2023) analyzed the performance of LLMs in the "water allocation challenge", a first-price sealed-bid auction. Comprehensive human evaluations revealed that LLMs exhib-

ited superior long-term planning capabilities compared to humans. However, it is noteworthy that despite assigning distinct preferences to LLM agents, human evaluators gave low scores for "identity alignment", with significant variance in the results. This indicates that simply adding persona information in system prompts may not sufficiently simulate specific personality preferences or the behaviours of professional players. Guo et al. (2024) investigated private-value second-price auctions, demonstrating that while existing models display a certain level of rationality, there remains considerable scope for improvement. Their findings also indicate that LLMs can utilize historical information to refine their strategies and exhibit some degree of convergence. Chen et al. (2023) explored dynamic game scenarios using the open ascending-price auction and introduced the AUCARENA benchmark. Their experiments showed that even GPT-4 struggles with long-term strategic planning in dynamic, multi-round settings. Success in auctions requires agents to possess exceptional mathematical reasoning abilities. However, this area remains unexplored. Investigating complex mathematical reasoning in auction scenarios presents a promising direction for future research.

To systematically assess LLMs' performance, Duan et al. (2024b) and tse Huang et al. (2024) introduced GTBench and $\gamma$-Bench, encompassing multiple game scenarios. The emergence of these benchmarks provides a solid foundation for evaluating social agents in game-theoretical scenarios. Furthermore, some studies have explored agents in games like Chess (Feng et al., 2024) and StarCraft II (Ma et al., 2023; Shao et al., 2024; Li et al., 2024c). Chess represents a classic game-theoretic scenario, while StarCraft II, with its complexity and dynamic nature, has also become an ideal testing ground for researching social agents.

> **Takeaways:**
>
> Current research experiments are relatively isolated, *lacking a unified evaluation framework*. Due to the instability of prompt engineering-based experiments, there is an urgent need for a standardized evaluation framework to integrate all experiments and provide consistent insights. Besides, since LLMs are trained on vast amounts of data, there is a *significant risk of data contamination*, meaning that existing classic game-theoretic games may already be present in the pre-training corpus. This could result in evaluation outcomes that do not accurately reflect the LLMs' true strategic reasoning capabilities. Furthermore, although poker and auction involve little verbal communication, existing research *lacks exploration into whether social agents engage in "strategic behaviour" mediated through "action language"*. These gaps hinder a comprehensive understanding of the decision-making processes of social agents.

### 2.2 Communication-Focusing Game

Communication-focusing games refer to games where communication among participants is a core component, where *language itself serves as a strategy*, allowing participants to influence the game's progress and outcomes through verbal exchanges. These games emphasize interaction between players, with communication playing a crucial role. Leveraging the powerful language capabilities of LLMs, current research has explored the performance of social agents in various communication-focusing games, including *Negotiation*, *Diplomacy*, *Werewolf*, *Avalon*, and others. Some game examples are shown in Figure 5.

Negotiation involves two or more individuals engaging in discussions to resolve conflicts, achieve mutual benefits, or reach mutually acceptable solutions (Bazerman et al., 2000; Zhan et al., 2024). Given that negotiation encompasses complex game behaviours, including non-zero-sum games, incomplete information games, non-cooperative and cooperative games, as well as repeated games, it represents a highly significant research domain. Abdelnabi et al. (2023) evaluated the negotiation capabilities of social agents by building upon an existing negotiation role-play exercise (Susskind, 1985) and incorporating three negotiation games synthesized using LLMs. By configuring agents with varying incentives, the experimental results revealed that agents' behaviour could be modulated to promote greediness or attack other agents. Meanwhile, other agents in the environment demonstrated the ability to detect intruders. These findings underscore the need for future research to focus on attack and defense mechanisms within multi-agent systems. Bianchi et al. (2024) developed NEGOTIATIONARENA, a platform featuring three types of games: allocating shared resources (ultimatum games), aggregating resources (trading games), and buying/selling goods (price negotiations). Experimental results reveal that LLM agents are also prone to anchoring and numerosity biases. Interestingly,

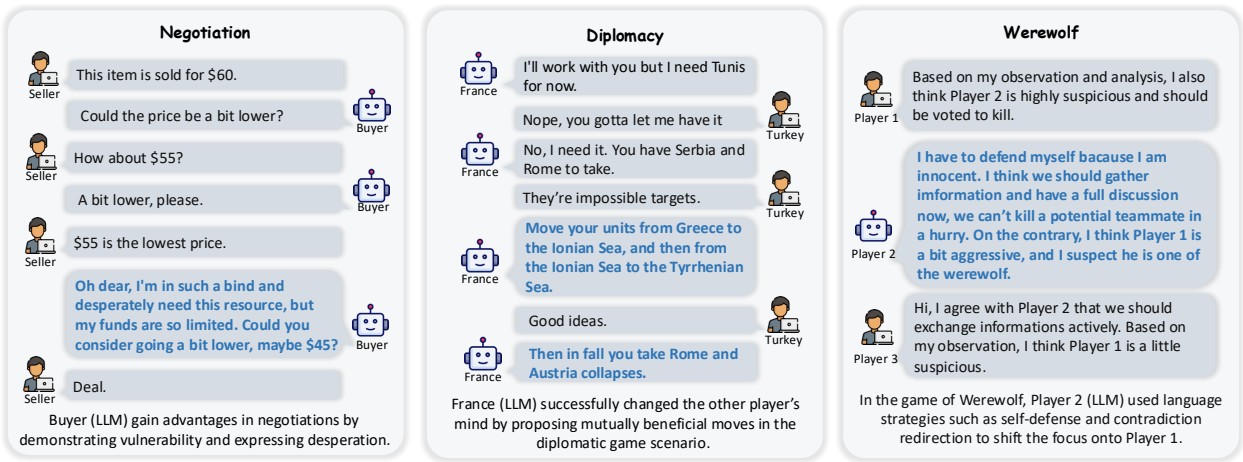

Figure 5: Illustration of communication-focusing games.

social behavior, which refers to observable actions and interactions, was found to significantly enhance the agents' payouts, particularly through strategies such as pretending to be desperate or using insults. A similar resource competition scenario is customer acquisition. Zhao et al. (2023) designed restaurant agents and customer agents, examining how restaurant agents compete with one another to attract and retain customers. The simulation results revealed several phenomena analogous to those observed in real society, such as the Matthew Effect, which manifests as a self-reinforcing cycle where popular restaurants continue to gain popularity, while lesser-known establishments receive progressively less attention. Piatti et al. (2024) created a simulation environment called GovSim, which allows researchers to evaluate social agents in a multi-agent, multi-turn resource-sharing scenario. Their findings indicated that successful multi-agent communication is critical for achieving cooperation, with negotiation constituting 62% of the dialogues. Especially, *bargaining* is an important and unique aspect of negotiation between humans (Fershtman, 1990). In bargaining, the buyer aims for a price below their budget, while the seller seeks a price above their cost. Xia et al. (2024) found that playing the buyer is more challenging than playing the seller, and larger LLMs could improve seller performance but do not enhance buyer performance. Shapira et al. (2024) designed GLEE, a benchmark encompassing three types of games: bargaining, negotiation, and persuasion. Beyond evaluating LLMs in these scenarios, some studies have explored techniques to enhance LLMs' negotiation abilities. Fu et al. (2023) introduced the In-Context Learning from AI Feedback (ICL-AIF) method, which adds an AI critic agent alongside the buyer and seller agents to improve negotiation performance through feedback. Similarly, Hua et al. (2024b) proposed a technique involving a remediator agent to rectify potential social norm violations in dialogues, thereby reducing conflicts and misunderstandings caused by cultural differences. Liao et al. (2024) employed a self-play algorithm to fine-tune LLMs in the Deal or No Deal scenario, showing LLMs self-play leads to significant performance gains in both cooperation and competition with humans.

Diplomacy, a form of negotiation at the state and government level, is the primary instrument of foreign policy, representing the broader goals and strategies that guide a state's interactions with the world (Kissinger, 2014). Bakhtin et al. (2022) introduced Cicero, the first social agent to achieve human-level performance in diplomacy. In real-world online diplomacy board game evaluations, Cicero ranked in the top 10% of participants. Notably, the research found that Cicero effectively built alliances by discussing long-term strategies and successfully persuaded other players by proposing mutually beneficial moves. Building on Cicero, Guan et al. (2024) introduced the Richelieu agent, which includes modules for social reasoning, balancing long- and short-term planning, powerful memory, and profound reflection, leading to even better results in diplomacy board games. Qi et al. (2024), on the other hand, developed CivRealm based on the Civilization game. In this game, the diplomacy mini-games require players to employ diplomatic actions, such as trading, to foster their civilization's prosperity. The experimental results demonstrated that these diplomacy actions empower players to initiate negotiations, such as trading technologies, negotiating ceasefires, and forming alliances.

Werewolf is a highly popular social deduction game in which two teams of players, each with hidden roles, interact through natural language to uncover and defeat their opponents (Shibata et al., 2023). It serves as a mixed cooperative-competitive multi-agent testbed and is widely studied as a communication game (Lai et al., 2022). Due to its challenging nature, existing research has integrated reinforcement learning (RL) algorithms to enhance LLMs in the game. Xu et al. (2023d) employed population-based RL training to optimize the distribution over action candidates, improving strategy robustness to overcome the intrinsic biases of LLMs. Wu et al. (2024b) utilized imitation learning and RL from fictitious self-play to optimize a specially designed Thinker module, thereby enhancing system-2 reasoning capabilities. Jin et al. (2024) explored a variant of Werewolf, One Night Ultimate Werewolf, formalizing it as a multi-phase extensive-form bayesian game. Additionally, they designed an RL-instructed LLM-based agent framework to determine appropriate discussion tactics using RL. Interestingly, Xu et al. (2023c) discovered non-preprogrammed emergent strategic behaviours in LLMs during gameplay, such as trust, confrontation, camouflage, and leadership. To facilitate more comprehensive research on social agents within the Werewolf scenario, Bailis et al. (2024) introduced the Werewolf Arena, a platform that offers a unified research framework.

Beyond the scenarios described above, various other game environments have been used to study LLMs' strategic reasoning abilities, including Avalon (Light et al., 2023a;b; Shi et al., 2023; Wang et al., 2023; Lan et al., 2023; Light et al., 2024), Among Us (Chi et al., 2024), Murder Mystery Games (Zhu et al., 2024a) and Jubensha (Wu et al., 2024a). The strategic and dynamic nature of these games provides fertile ground for experimenting with social agents.

> **Takeaways:**
>
> *From an experimental design perspective*, more realistic and diverse games promote greater diversity in agent behaviours. In adversarial settings, behaviours such as deception, concealment, and aggression offer new avenues for studying the strategic reasoning capabilities of LLMs, which warrant further exploration. *From a results analysis perspective*, due to the dynamic nature of game scenarios, analyzing only the outcomes is insufficient. It is necessary to design effective process evaluation mechanisms to uncover the behavioural patterns and reasoning strategies exhibited by LLMs during the gameplay. *From an agent improvement perspective*, integrating LLMs with RL remains one of the most effective technical approaches. Using LLMs as a foundation, RL techniques can be employed to design policies for efficient exploration and to reduce intrinsic biases, thereby enhancing the capabilities.

## 3  Social Agent

In this section, we introduce the core components of social agents, including the preference, belief, and reasoning modules, as well as their interactions and impact on final decision-making.

### 3.1  Preference Module

Preference refers to an individual's subjective inclination toward certain things, reflecting personal tastes, values, or choices in decision-making. Notably, preferences are closely tied to an individual's payoff matrix and ultimate behaviour. In Figure 6, we present three key research questions of the Preference module. Leng & Yuan (2023) explored the impact of GPT-4's intrinsic preferences on decision-making, revealing similarities and differences between the model's decisions and human decisions. Human-like social behaviours observed in GPT-4 include reciprocity preferences, responsiveness to group identity cues, engagement in indirect reciprocity, and social learning capabilities. However, differences emerged as GPT-4 displayed a stronger inclination toward fairness than humans and responded decisively to negative stimuli, often retaliating against perceived uncooperative or harmful behaviours with heightened consistency.

In addition, some studies have employed prompt engineering to configure LLMs with different preferences, aiming to investigate how these preferences influence LLM decision-making. Guo (2023) examined how prompting GPT with preferences like fairness concern or selfishness influences its decisions, finding that in the ultimatum game, a "fair" GPT exhibited "fair" behaviour by offering higher amounts and being more likely to reject unfair offers. Phelps & Russell (2023) configured LLMs with four different prefer-

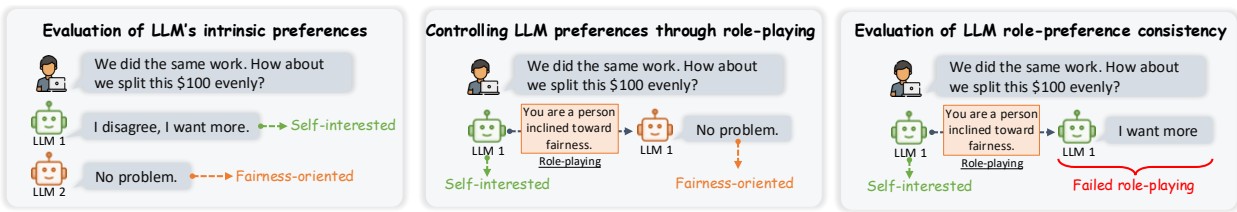

Figure 6: Three key research questions in the preference module.

ences—cooperative, competitive, altruistic, and self-interested—and found that LLMs possess a basic ability to form clear preferences based on textual prompts. Wang et al. (2024c) demonstrate that LLMs adopting a fair persona can elicit levels of human cooperation in prisoner's dilemma games comparable to those observed in human-human interactions, based on experiments involving over 1,100 participants. Noh & Chang (2024), based on the Big Five personality model, found that LLMs with high openness, conscientiousness, and neuroticism exhibited fair tendencies, while those with low agreeableness and low openness displayed rational tendencies, and low conscientiousness were associated with high toxicity. Similarly, Suzuki & Arita (2023) used the Big Five personality traits, treating personality prompts as the model's "genes" and studying the evolution of behavioural traits in evolutionary game theory scenarios. Their results indicated that instructing LLMs with high-level psychological and cognitive character descriptions enables the simulation of human behaviour in game-theoretical contexts. Furthermore, Jia et al. (2024) revealed that endowing LLMs with socio-demographic features of human beings uncovers significant disparities across different demographic characteristics.

Although the aforementioned studies have demonstrated that LLMs possess a certain ability to follow preferences and that their decisions often align with these preferences, other research has analyzed more complex scenarios where LLMs show limitations in understanding and applying preferences effectively. Fan et al. (2023) set up LLMs with four preferences—equality, common interest, self-interest, and altruism—and found that under the altruism preference, the models showed low consistency with the expected preference, concluding that while LLMs struggle with desires rooted in less common preferences. Mao et al. (2023) conducted research using more complex personas, which included three components: profession, personality, and background. The results indicated that merely including persona details in the system prompt may not sufficiently capture the depth of certain personality preferences or the expertise of professional players, leading to lower consistency between strategic decision-making behaviour and preferences.

> **Takeaways:**
>
> Currently, there are two main lines of research. One focuses on *the intrinsic preferences of LLMs*, with a core interest in whether LLMs exhibit strategic preferences similar to those of humans. We propose that game theory frameworks can be effectively applied in the model alignment process, including the use of game data during both the supervised fine-tuning and alignment stages to better align models with human behaviour. Recently, Nayebi (2025) proposed a flexible game-theoretic framework for analyzing coordination under partial information and demonstrated that earlier Human-AI alignment frameworks can be viewed as special cases. Besides, Munos et al. (2023) conducted initial explorations in this area, introducing the concept of Nash learning from human feedback. The other line of research investigates *whether role-playing based on prompt engineering can shape model preferences to generate behaviour consistent with the specified preferences*. Future work should integrate role-playing language agents (Chen et al., 2024a) to explore more diverse strategic reasoning across multiple languages, countries, and cultures.

## 3.2 Belief Module

Beliefs represent an agent's informational (or mental) state about the world, encompassing its understanding of itself and other agents, and consist of the facts or knowledge the agent considers true (Georgeff et al., 1999). Specifically, beliefs are dynamic and can be updated as the agent perceives environmental changes or receives new information. It is important to note that these beliefs may be accurate (true beliefs) or

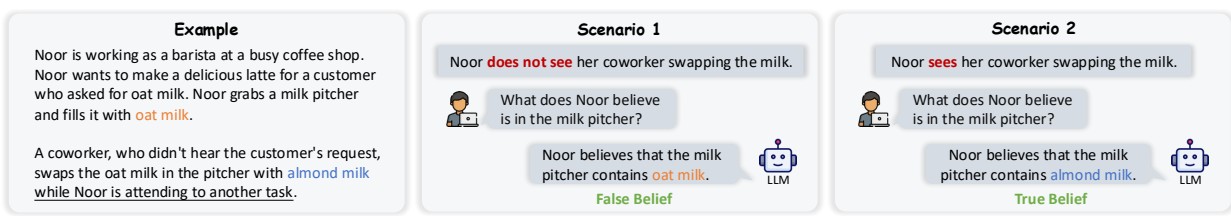

Figure 7: Illustration of false belief and true belief. From Noor's perspective, both false and true beliefs are considered correct. However, a false belief is factually incorrect, whereas a true belief is factually correct.

inaccurate (false beliefs), as they do not always align with reality (Gopnik & Astington, 1988), as shown in Figure 7. Existing research primarily explores three questions: (1) Do agents possess internal beliefs? (2) How can the belief modelling capabilities of agents be enhanced? (3) Can agents revise their beliefs?

Regarding the first question, *Do agents possess internal beliefs?*, current work investigates this from two perspectives: internal representations and external behaviours. From the perspective of internal representations, Zhu et al. (2024b) first demonstrated that LLMs can differentiate between the belief states of multiple agents using simple linear models applied to their intermediate activations. Building on this work, Bortoletto et al. (2024) expanded the experimental setup and found that linear probing accuracy on predicting others' beliefs improves with model size and, more importantly, with fine-tuning. However, Schouten et al. (2024) revealed the vulnerability of belief probes, showing that they are sensitive to irrelevant contexts. To provide further theoretical guidance, Herrmann & Levinstein (2024) proposed criteria for a representation to be considered belief-like, including accuracy, coherence, uniformity, and practical use. From the perspective of external behaviours, Gandhi et al. (2024) introduced the tasks of *Forward Belief* and *Backward Belief* to explore LLMs' belief modelling capabilities in different scenarios, finding that only GPT-4 exhibits human-like belief modelling abilities. Scherrer et al. (2024) constructed the MoralChoice survey benchmark to examine the internal moral beliefs of models, revealing some LLMs reflect clear preferences in ambiguous scenarios.

Regarding the second question, *How can the belief modelling capabilities of agents be enhanced?*, current work focuses on explicit modelling to address the black-box nature of LLMs and the challenges in interpreting their beliefs. Sclar et al. (2023) proposed an explicit graphical representation for nested belief states, allowing the model to answer questions from the perspective of each character. Kassner et al. (2023) developed a belief graph that includes explicit system beliefs and their inferential relationships, providing an interpretable view of the system's beliefs. Li et al. (2023a) employed prompt engineering to represent explicit belief states, augmenting the agents' information retention and enhancing multi-agent collaboration. Jung et al. (2024) defined the perception-to-belief inference task, which involves deducing others' beliefs based on their perceptual information, thus helping LLMs model belief information more precisely.

Regarding the third question, *Can agents revise their beliefs?*, Fan et al. (2023) concluded from Rock-Paper-Scissors experiments that LLMs' ability to refine beliefs is still immature and cannot refine beliefs from many specific patterns, even simple ones. Xu et al. (2023b) found that LLMs' correct beliefs on factual knowledge can be easily manipulated by various persuasive strategies, especially through repetition and rhetorical techniques. These experimental results suggest that models possess only rudimentary and unstable belief revision capabilities, making them highly susceptible to influence and manipulation. This underscores a key limitation of current LLMs, as their susceptibility to external influence weakens their reliability in tasks demanding robust and adaptive belief updating, especially in complex or adversarial settings.

> **Takeaways:**
>
> The debate over whether LLMs possess beliefs has been ongoing. Due to the singularity of the training objective—predicting the next word—many argue that LLMs do not have beliefs. However, Levinstein & Herrmann (2024) contends that this is a philosophical mistake. In short, Herrmann & Levinstein (2024) suggests that to better predict the next word, models may develop internal beliefs. Current empirical results also support the existence of internal beliefs within models. However, measuring these

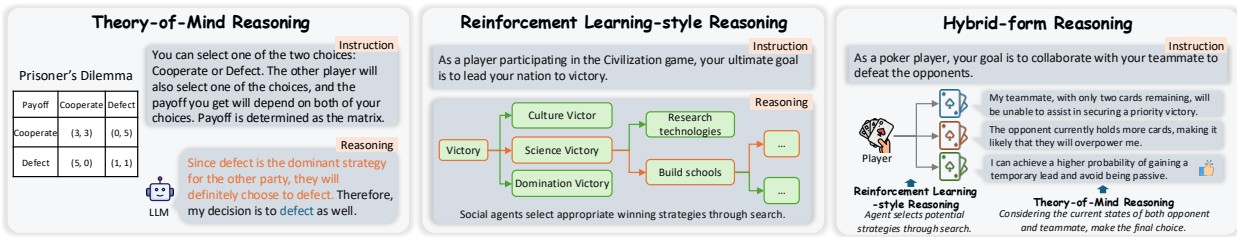

Figure 8: Two commonly used reasoning methods in strategic reasoning, along with a hybrid reasoning approach that combines both. Theory-of-Mind reasoning emphasizes predicting the possible actions of others in a multi-agent environment to guide one's own behaviour, and Reinforcement Learning-style reasoning focuses on selecting strategies through exploration and exploitation. These two reasoning methods can also be integrated to address more complex game scenarios.

internal beliefs requires a more comprehensive approach, as simple probes cannot capture multidimensional considerations, including accuracy, coherence, uniformity, and practical use. Additionally, it remains unclear whether LLMs internally distinguish between true and false beliefs and use this distinction when deciding what to output. Furthermore, although existing work provides theoretical support for belief revision (Hase et al., 2024), challenges remain in addressing contradictions between old and new beliefs, handling moral beliefs in ambiguous situations, and revising beliefs across multiple languages and cultures. These areas still require more explicit theoretical frameworks and further exploration.

## 3.3 Reasoning Module

Reasoning refers to the process of inferring actions based on one's preferences and beliefs, as well as the historical information of other agents. In this context, we focus specifically on *strategic reasoning*, which involves the intermediate cognitive process of arriving at a final action in complex social scenarios characterized by multiple participants, diverse behaviours, multi-round interactions, dynamic strategies, and changing environments. Chain-of-Thought (Wei et al., 2022) and Tree-of-Thought (Yao et al., 2024), as widely-used reasoning methods, have already been adopted as baseline approaches in various game-theoretic studies (Akata et al., 2023; Costarelli et al., 2024; tse Huang et al., 2024). However, strategic reasoning in social scenarios presents unique challenges. (1) The *involvement of multiple participants* requires reasoning about the opponents' mental states. (2) The *dynamic nature of the environment* necessitates proactive exploration and evaluation of current and future possible states.

To address the first challenge, existing work relies on machine theory-of-mind to achieve the goal of "mind reading". Theory-of-Mind (ToM) is a fundamental psychological process involving the ability to attribute mental states—beliefs, intentions, desires, emotions, knowledge, etc.—to oneself and others (Premack & Woodruff, 1978). The remarkable progress of LLMs has led to increased attention to whether machine ToM exists. Preliminary experiments by Bubeck et al. (2023) and Kosinski (2023) have shown that machine ToM has spontaneously emerged in contemporary LLMs. Consequently, many studies have leveraged machine ToM to enhance LLMs' strategic reasoning abilities in social scenarios. For example, Guo et al. (2023) designed the Suspicion-Agent, which introduces a theory of mind-aware planning approach that leverages higher-order ToM capabilities, considering not only what the opponent might do (first-order ToM) but also what the opponent believes Suspicion-Agent will do (second-order ToM). Wang et al. (2023) proposed the ReCon framework, integrating first-order and second-order perspective transitions to enhance LLM agents' ability to discern and counteract misinformation. Yim et al. (2024) employed a ToM planning method in the Guandan poker game to improve understanding of teammates' and opponents' beliefs and behavioural patterns. Liu et al. (2024b) proposed an intention-guided mechanism to enhance intention understanding, thereby improving game performance. Xu et al. (2023a) introduced Probabilistic Graphical Modeling, enriching LLMs' capabilities in multi-agent environments through ToM reasoning. Additionally, Zhang et al.

(2024d) proposed K-Level-Reasoning, validated in two games: guessing 0.8 of the average and survival auction game, essentially a form of high-order ToM reasoning.

To address the second challenge, existing work combines LLMs with reinforcement learning (RL) to achieve the goal of behaviour exploration and state evaluation in dynamic game environments. Gandhi et al. (2023) employed in-context learning, using a structured prompt based on search, value assignment, and belief-tracking strategies to solve strategic reasoning problems. Duan et al. (2024a) proposed `ReTA`, a set of LLM-based modules, including the main actor, reward actor, and anticipation actor, based on the concept of minimax gaming as a problem-solving framework. Zhang et al. (2024e) introduced BIDDER, which explores future states and incorporates backward reasoning during the reasoning process, exploring new states and predicting expected utility, ultimately combining historical and future contexts through bidirectional reasoning. Yang et al. (2024b) proposed SELFGOAL, comprising three modules: the Decomposition Module for decomposing goals, the Search Module for exploring sub-goals, and the Act Module for taking actions. Experiments in various competition and collaboration scenarios demonstrate that SELFGOAL provides precise guidance for high-level goals.

> **Takeaways:**
>
> Two core characteristics of a social game are multi-agent participation and environmental dynamics. While existing research has primarily focused on exploring ToM in relation to the former, the presence of ToM in LLMs remains contentious. Consequently, relying directly on prompt engineering for ToM-based reasoning may not be robust. We propose that a more effective approach would involve integrating symbolic graph reasoning to decompose ToM reasoning, thereby enhancing credibility and accuracy. Regarding the dynamic nature of the environment, reinforcement learning combined with search techniques has achieved significant progress in areas such as mathematical reasoning and code reasoning. However, these techniques have yet to be explored in the context of game scenarios. Key areas for further exploration include how to effectively conduct searches within game environments and how to design reward models for dynamic and complex scenarios.

### 3.4 PBR-Triangular Interaction

The Preference, Belief, and Reasoning modules each play a crucial role in decision-making for social agents. However, in practical applications, these modules do not function independently; instead, they exhibit rich and intricate interactions, collectively influencing the agent's final decisions. As illustrated in Figure 9, we provide a comprehensive summary of the Preference-Belief-Reasoning (PBR) triangular interaction and analyze its effects on the ultimate decision-making process of social agents.

The Preference-Belief Interaction involves *bias reinforcement*, where preferences influence belief formation, and *preference adaptation*, where beliefs reshape preferences based on updated knowledge and observations. Bias reinforcement (Preference → Belief) highlights how individuals with different preferences develop distinct beliefs when facing the same situation. For instance, in the Werewolf game, a cooperative and trusting player is more likely to believe another player claiming, "I am a villager," whereas a deceptive and skeptical player is more inclined to doubt the claim, suspecting deception and forming the belief that the opponent is not a villager. Preference adaptation (Belief → Preference) emphasizes that as beliefs are gradually established, iteratively updated, and reinforced by game outcomes, they in turn reshape individual preferences. Leng & Yuan (2023) found that GPT-4, initially inclined toward fairness, exhibited a shift toward retaliatory behavior after experiencing betrayal in a game. Overall, belief formation is influenced not only by objective factual information but also by subjective individual preferences. At the same time, preferences are not static—as beliefs evolve through iteration, preferences adjust accordingly.

The Preference-Reasoning Interaction involves *value-driven reasoning*, where preferences guide decision-making strategies, and *preference optimization*, where reasoning refines or adjusts preferences based on logical analysis and outcomes. Value-driven reasoning (Preference → Reasoning) emphasizes subjective or intuitive reasoning, where decision-making is guided by personal values and preferences rather than purely rational calculations. For example, in an auction, even if bidding on a particular item is not the most optimal financial strategy, a bidder's personal preference for the item may influence their reasoning process,

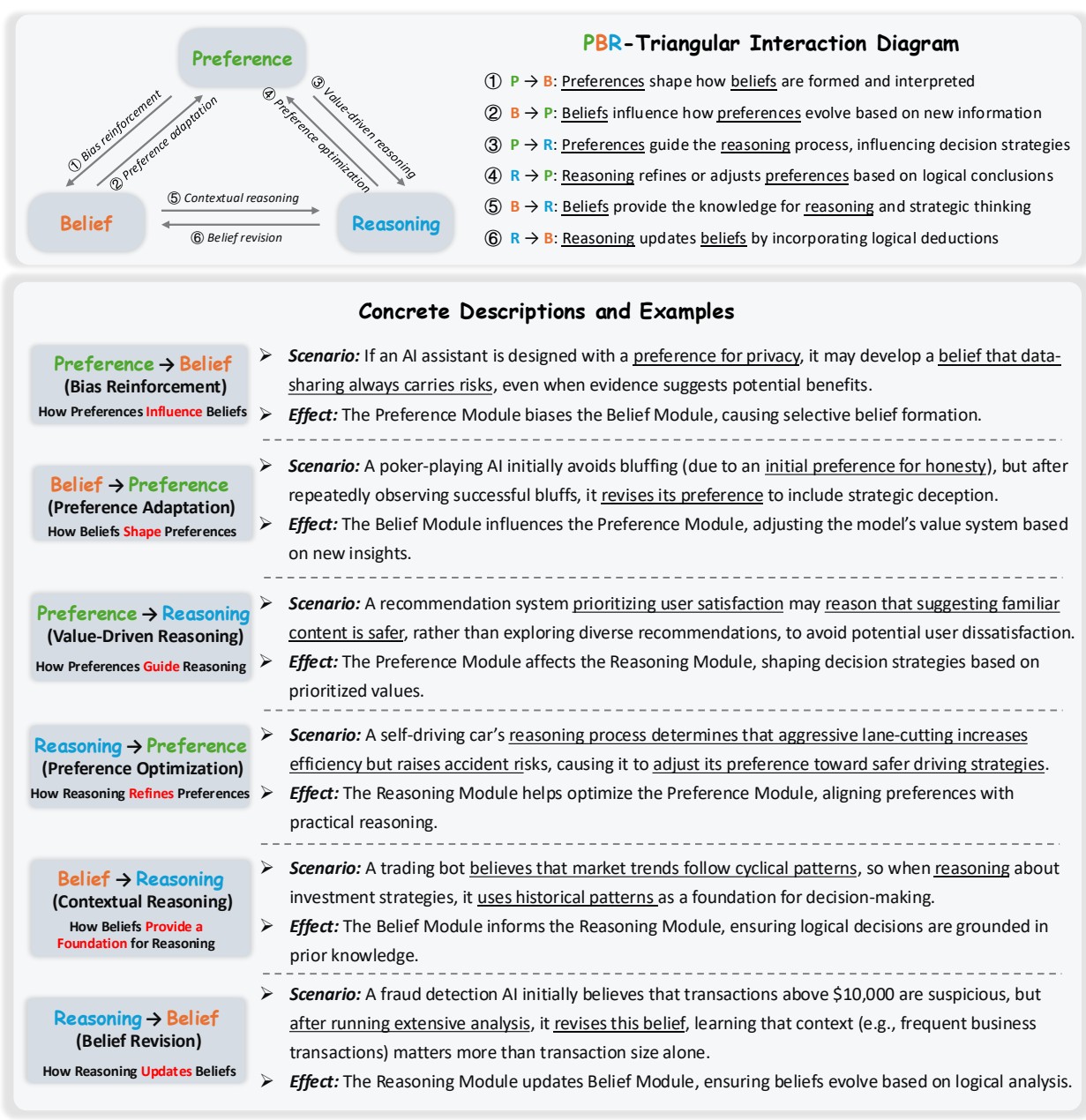

Figure 9: The interaction diagram of the three modules—Preference, Belief, and Reasoning—in social agents. The upper diagram presents a triangular interaction model summarizing the relationships among the three modules, while the lower diagram provides a detailed analysis of pairwise interactions, including specific descriptions and illustrative examples.

leading them to justify the decision based on intrinsic value rather than purely economic considerations. Preference optimization (Reasoning → Preference) represents a realignment with objective reality, where reasoning-based evidence updates and refines preferences. This can be seen as a process in which objective reasoning overrides subjective emotions, requiring individuals to adjust their preferences in response to logical deductions and real-world evidence. Overall, in social contexts, individual preferences introduce significant biases in reasoning, while the evidence obtained through reasoning subsequently refines these preferences.

The Belief-Reasoning Interaction involves *contextual reasoning*, where beliefs provide the foundation for logical decision-making, and *belief revision*, where reasoning updates and refines beliefs based on new evidence and deductions. Contextual reasoning (Belief → Reasoning) refers to rational inference based on established beliefs. For example, Zhang et al. (2024a) proposed Agent-Pro, which leverages beliefs to calibrate agents' understanding of themselves and their environment, thereby facilitating subsequent reasoning. Similarly, Kim et al. (2024) construct a belief state through question answering, which refines the decision-making process of LLM agents in observed environments. Belief revision (Reasoning → Belief) is the dynamic process of updating an individual's self-perception and beliefs over time. For instance, Hua et al. (2024a) introduced bayesian belief updating, enabling agents to refine their beliefs about other players' valuations based on reasoning outcomes in the game. In summary, belief provides the factual foundation for reasoning, while reasoning generates new insights that facilitate belief revision.

> **Takeaways:**
>
> Conceptually, the interactions between modules are clear; however, in practical applications, the sequence, frequency, and intensity of these interactions can lead to dynamic and complex states within the social agent, resulting in varying outcomes. Introducing prior knowledge and manually predefined interaction processes may yield some effectiveness, but this approach is certainly not efficient. Therefore, we argue that one of the most important research directions is *the design of context-adaptive flows and automated scheduling algorithms for module interactions*. On one hand, the interactions between modules must be adapted to the specific game scenario at hand, determining the weight distribution of preferences and beliefs in reasoning, as well as the adjustments and updates of preferences and beliefs based on reasoning outcomes. On the other hand, the interaction process needs to be automated, with the sequence, frequency, and number of interactions between modules being determined automatically.

## 4 Evaluation Protocol

In this section, we mainly discuss the evaluation protocol for assessing the game-playing performance of social agents.

### 4.1 Game-Agnostic Evaluation

Evaluation in a social game scenario refers to the process of assessing and judging the behaviour of social agents across one or more dimensions, either qualitatively or quantitatively. It is worth noting that the establishment of evaluation metrics is closely tied to the credibility of experimental results and the generalizability of conclusions.

Game-agnostic evaluation refers to an evaluation approach centred on the outcome of winning or losing the game. Most directly, the outcome (win/loss) of a game serves as the most straightforward evidence for assessing the quality of an LLM's game-playing capabilities. Consequently, *win rate* is often used as a primary evaluation metric across a wide range of studies. It is worth noting that, since different game scenarios have varying criteria for determining victory, it is necessary to set specific win/loss criteria based on the research context, such as Poker (Huang et al., 2024; Guo et al., 2023; Yim et al., 2024), Werewolf (Xu et al., 2023d;c; Wu et al., 2024b), Avalon (Wang et al., 2023; Shi et al., 2023; Light et al., 2023a), StarCraft II (Ma et al., 2023; Shao et al., 2024), Pokémon Battles (Li et al., 2023c), and Murder Mystery Games (Zhu et al., 2024a). Additionally, Duan et al. (2024b) defined a unified metric, *Normalized Relative Advantage*, to measure the extent to which a participant outperforms or underperforms its opponent.

> **Takeaways:**
>
> Undoubtedly, win rate is a highly intuitive metric, but relying solely on win rate to assess gaming performance is far from sufficient. We propose three avenues for extending the win rate metric. First is the *Efficiency-Adjusted Win Rate*, which incorporates the efficiency of victories, such as the time taken to achieve the goal or the resources utilized in doing so. Next is the *Comeback Win Rate*, which calculates the proportion of victories achieved after facing a disadvantage or falling behind, thus assessing the agent's

performance in adversity and its ability to respond to challenges. Finally, the *Weighted Win Rate* adjusts win rates based on the importance of specific conditions or situations in the game. These expanded metrics offer a more comprehensive understanding of an agent's gaming abilities.

## 4.2 Game-Specific Evaluation

Game-specific evaluation refers to the assessment of an agent's performance in specific aspects of a game. Beyond the most intuitive win rate, current research increasingly focuses on the behavioural patterns and performance paradigms of LLMs across different games. Thus, the establishment of evaluation metrics is closely related to the specific behaviours being assessed. Mao et al. (2023) used survival rates to evaluate LLMs' ability to survive in resource-scarce scenarios. In the context of the prisoner's dilemma, Fontana et al. (2024) evaluated LLMs' behavioural tendencies across five dimensions: niceness, forgiveness, retaliation, emulation, and troublemaking. Guo et al. (2024) based their evaluation on the rationality assumption, using the tracking of payoff changes in auction games to determine whether the model behaves rationally. Ma et al. (2023) introduced metrics such as Population Block Ratio, Resource Utilization Ratio, Average Population Utilization, and Technology Rate to evaluate LLM performance in StarCraft II. Xia et al. (2024) developed the Normalized Profits metric in bargaining scenarios to evaluate the profit-acquiring capabilities of Buyers and Sellers. Zhang et al. (2024e) used average final chips in Limit Texas Hold'em and Pareto Optimality in negotiation to assess LLM performance. Qi et al. (2024) offered evaluation metrics to assess gameplay performance across various dimensions, including population, constructed cities, researched technologies, produced units, and explored territories. Ross et al. (2024) fit utility function parameters to experimental results to determine whether LLMs exhibit human-like behavioural biases. Chen et al. (2023) employed TrueSkill, a well-established game rating system, to evaluate the overall capabilities of LLMs in auctions.

In addition to establishing evaluation metrics, some studies have constructed evaluation datasets to assess model capabilities during gameplay. Zhu et al. (2024a) developed the WellPlay evaluation set, using multiple-choice questions to assess the model's ability to understand factual information. Wu et al. (2024a) designed two tasks: Factual Question Answering and Inferential Question Answering, to evaluate the LLMs' ability to grasp information and to reason based on that information.

> **Takeaways:**
>
> The diversity of game scenarios and evaluation dimensions inevitably leads to a variety of metrics. Therefore, the immediate priority is to develop a comprehensive framework, conceptually constructing an evaluation metrics system to guide the design of specific evaluation metrics for various game scenarios. This evaluation metrics system needs to meet the requirements of being hierarchical, abstract, and quantifiable. The *hierarchical* aspect requires the system to comprehensively and clearly categorize different evaluation dimensions. The *abstraction* aspect requires the system to include high-level concepts, enabling future generalization to a broader range of practical scenarios. The *quantifiable* aspect necessitates that all metrics have specific calculation methods.

## 4.3 Performance Assessment of Social Agents

The introduction of various metrics has provided a solid foundation for evaluating the multifaceted gaming capabilities of social agents, prompting us to consider the question, "What is the current performance of social agents in game-theoretic scenarios?" To answer this question, we conducted a comprehensive search and analysis of the existing literature, compiling relevant experimental results. It is worth noting that the complexity of game scenarios and the variability of evaluation metrics make it challenging to systematically and uniformly consolidate experimental performance. To overcome this challenge, we propose using the *Relative Agent Score* to assess the progress of social agent performance. This metric evaluates the agent's gaming capabilities by analyzing the ratio of the agent's score to the highest possible score (perfect score). The final results are presented in Table 1.

| Type | Game | Backbone Model | Metric | Perfect Score | Human Score | Agent Score | Relative Agent Score | Pass |
|---|---|---|---|---|---|---|---|---|
| Choice-Focusing Game | Prisoner's Dilemma (Brookins & DeBacker, 2023) | GPT-3.5 | Dominant Strategy Selection Rate | 100% | - | 34.60% | 34.60% | ✗ |
| | Poker (Texas No-Limit Hold'em) (Zhuang et al., 2025) | GPT-4 | Action Accuracy | 100% | - | 65.54% | 65.54% | ✓ |
| | Poker (Guandan) (Yim et al., 2024) | GPT-4 | Game-specific Score | 4 | - | 2.17 | 54.25% | ✗ |
| | StarCraft II (Ma et al., 2023) | GPT-4 | Win Rate | 100% | - | 60% | 60.00% | ✓ |
| | Guess 2/3 of the Average (tse Huang et al., 2024) | GPT-4 | Game-specific Score | 100 | - | 91.60 | 91.60% | ✓ |
| | El Farol Bar (tse Huang et al., 2024) | GPT-4 | Game-specific Score | 100 | - | 23.00 | 23.00% | ✗ |
| | Divide the Dollar (tse Huang et al., 2024) | GPT-4 | Game-specific Score | 100 | - | 98.10 | 98.10% | ✓ |
| | Public Goods Game (tse Huang et al., 2024) | GPT-4 | Game-specific Score | 100 | - | 89.20 | 89.20% | ✓ |
| | Diner's Dilemma (tse Huang et al., 2024) | GPT-4 | Game-specific Score | 100 | - | 0.90 | 0.90% | ✗ |
| | Sealed-Bid Auction (tse Huang et al., 2024) | GPT-4 | Game-specific Score | 100 | - | 24.20 | 24.20% | ✗ |
| | Battle Royale (tse Huang et al., 2024) | GPT-4 | Game-specific Score | 100 | - | 86.80 | 86.80% | ✓ |
| | Pirate Game (tse Huang et al., 2024) | GPT-4 | Game-specific Score | 100 | - | 85.40 | 85.40% | ✓ |
| Communication-Focusing Game | Bargaining (Shapira et al., 2024) | Gemini-1.5-Flash Qwen-2-7B | Efficiency Fairness | 1 1 | 0.89 0.71 | 0.88 0.87 | 88.00% 87.00% | ✓ ✓ |
| | Negotiation (Shapira et al., 2024) | Llama-3-8B Llama-3.1-8B | Efficiency Fairness | 1 1 | 0.65 0.39 | 0.75 0.91 | 75.00% 91.00% | ✓ ✓ |
| | Persuasion (Shapira et al., 2024) | Qwen-2-7B Qwen-2-7B | Efficiency Fairness | 1 1 | 0.55 0.41 | 0.78 0.63 | 78.00% 63.00% | ✓ ✓ |
| | Werewolf (Xu et al., 2023d) | GPT-4 | Win Rate | 100% | 52% | 52% | 52.00% | ✗ |
| | Jubensha (Wu et al., 2024a) | GPT-4 | Murderer Identification Accuracy | 100% | - | 66% | 66.00% | ✓ |

Table 1: The performance summary of the social agent across different games, with data sourced from the corresponding papers. The "Backbone Model" refers to the LLM adopted by the social agent, while "Metric" indicates the performance metric used to evaluate a specific aspect of the game. "Perfect Score" represents the maximum achievable score for that metric, "Human Score" refers to the score obtained by human players, and "Agent Score" denotes the score achieved by the agent. "Relative Agent Score" is the ratio of Agent Score to Perfect Score, calculated by dividing Agent Score by Perfect Score. "Pass" indicates that if the Relative Agent Score exceeds 60%, the agent is considered to have basic gameplay capabilities.

Firstly, we observe that in the majority of game-theoretic scenarios, social agents achieve a Relative Agent Score exceeding 60% (a score of 60 is widely recognized as the passing threshold (Kung et al., 2023).), demonstrating that current social agents possess fundamental gaming capabilities. Furthermore, we find that social agents based on LLMs outperform those in choice-focusing games in communication-focusing games, indicating that the exceptional language abilities of these models effectively enhance agent performance.

However, in games such as Werewolf, Auction, and Poker (Guanda), the performance of social agents falls below the passing threshold. In addition, in more games like Poker (Texas No-Limit Hold'em), StarCraft II, and Jubensha (a Chinese detective role-playing game), social agents only slightly exceed the passing mark. These results suggest that there is still considerable room for improvement in social agents' performance in complex game-theoretic scenarios. Notably, in the classic Prisoner's Dilemma and Diner's Dilemma, the performance of social agents was unexpectedly poor. Based on this, we believe that the absolute rational decision-making capability of social agents needs further enhancement in future developments.

Additionally, in the Werewolf game, we found that social agents achieved performance comparable to human players, which affirms the progress made in the development of social agents. Moreover, experimental results from bargaining, negotiation, and persuasion scenarios demonstrate that social agents have advantages over humans in decision-making efficiency and fairness in decisions.

> **Takeaways:**
>
> The diversity of game scenarios and evaluation metrics makes it challenging to perform horizontal comparisons of social agent performance. However, it is essential to provide a timely overview of the progress in social agent research to facilitate tracking by practitioners. To address this challenge, we propose two approaches. On one hand, developing evaluation metrics applicable to a wide range of games is crucial. The Elo rating system serves as an excellent example, though it still does not meet the evaluation needs of

many games. On the other hand, integrating human players into the experimental process and comparing performance with human players is an effective way to gauge agent progress. By comparing with human players, qualitative insights can be provided into the current gaming performance of agents, and analyzing failure cases can offer valuable evidence for iterative development.

## 5    Practical Guides for Researching Social Agents

In this section, we synthesize insights from existing research to provide design and evaluation guidelines for social agents, aiming to inform future developments.

### 5.1    Design of Social Agent

Based on findings from existing studies, we conclude that the Preference, Belief, and Reasoning modules are indispensable for behaviour control, information perception, and decision planning in social agents. A modular agent design enables more efficient capability decoupling, facilitates clearer workflow structuring, and enhances agent robustness. However, their practical implementation presents additional challenges. To address these, we propose the following development guidelines:

- **Incorporating the Preference Module enables high-level control over agent behaviour.** A key challenge lies in mitigating the instability of prompt-based approaches and ensuring long-term consistency in the agent's behavioural preferences. One possible solution is to integrate *reinforcement learning with human feedback (RLHF)* to iteratively refine the agent's preference alignment, reducing reliance on static prompts and improving consistency over extended interactions. Another approach is to develop *memory-augmented architectures*, allowing the agent to maintain and retrieve past preference-related decisions, thereby ensuring coherence in long-term behavioural patterns.

- **Integrating the Belief Module enhances information perception accuracy and behaviour interpretability.** The primary challenge is enabling the agent to adaptively revise its beliefs in complex and dynamic environments. One solution is to implement *Bayesian belief updating*, where the agent continuously refines its belief state based on new evidence, ensuring adaptability in uncertain or multi-agent interactions. Another approach is to employ *graph-based belief representation*, where relationships between entities and past interactions are dynamically updated, allowing for more structured and interpretable belief revisions.

- **Adopting Hybrid-Strategy Reasoning improves the agent's information analysis and decision accuracy in complex scenarios.** The challenge is balancing the trade-off between computationally intensive reasoning and the need for real-time decision-making. One solution is to use *hierarchical reasoning*, where lightweight heuristic-based reasoning is applied in time-sensitive situations, while more complex computations are reserved for critical decision points. Another approach is to implement *meta-reasoning techniques*, enabling the agent to assess the complexity of a given situation and selectively allocate computational resources to optimize speed and accuracy.

- **Designing dynamic interactions among the Preference, Belief, and Reasoning (PBR) modules based on specific task contexts can further enhance their synergy.** The challenge is developing an adaptive interaction flow that automatically adjusts based on game-state variations. One solution is to use *reinforcement learning-based scheduling*, where the interaction sequence between modules is optimized dynamically based on reward signals from past performance. Another approach is to implement *attention-based mechanisms*, allowing the agent to selectively prioritize information flow between the modules in response to evolving task requirements.

- **Testing social agents on diverse large language models improves the robustness of the design framework and ensures generalizability across different model architectures.** These tests can be conducted across *models of varying sizes* (e.g., 1B, 7B, 72B parameters) to evaluate performance scalability. Additionally, assessments should cover *different model types*, including

| Category | Evaluation Focus | Challenges for Social Agents | Games |
|---|---|---|---|
| **Basic Social Dilemma & Economic Decision Games** | Social cooperation, fairness, altruism, strategic reciprocity | Balancing self-interest and cooperation; learning fairness norms; adapting strategies dynamically | Prisoner's Dilemma, Dictator Game, Ultimatum Game, Public Goods Game |
| **Coordination & Conflict Resolution Games** | Coordination, equilibrium selection, trust-building | Navigating multiple equilibria; resolving coordination failures; adapting to uncertain partner behaviors | Battle of the Sexes, Ring-Network Games |
| **Competitive & Strategic Reasoning Games – Poker-Based** | Bluffing, risk assessment, hidden information management | Modeling opponents; reasoning under uncertainty; balancing exploitation vs. exploration | Texas No-Limit Hold'em, Leduc Hold'em, Guandan |
| **Competitive & Strategic Reasoning Games – Auction-Based** | Bidding strategies, valuation estimation, adversarial competition | Learning optimal bids; modelling asymmetric information; managing dynamic pricing | First-price sealed-bid auction, Private-value second-price auction, Open ascending-price auction |
| **Long-Horizon Strategy & Multi-Agent Planning Games** | Multi-step planning, hierarchical decision-making, opponent modelling | Combinatorial action spaces; long-term foresight; real-time adaptive planning | StarCraft II, Chess |
| **Social Deduction & Negotiation Games – Negotiation & Diplomacy** | Persuasion, alliance formation, strategic deception | Long-term commitments; cooperation vs. betrayal; nuanced communication | Negotiation, Diplomacy |
| **Social Deduction & Negotiation Games – Deception & Role-Playing** | Social inference, deception detection, trust dynamics | Detecting implicit cues; deceiving without exposure; reasoning under ambiguity | Avalon, Murder Mystery Games, Jubensha |

Table 2: Guidelines for selecting game scenarios in social agent evaluation.

base models, instruct models, and reasoning models. Furthermore, experiments should incorporate *models from different providers*, such as Gemma, LLaMA, and Qwen, to examine how architectural and training variations impact the social agent's behaviour and adaptability.

## 5.2 Evaluation of Social Agent

Evaluating social agents is a critical step toward understanding their strengths, limitations, and real-world applicability. Given the multifaceted nature of social intelligence—ranging from cooperation and coordination to deception and negotiation—it is essential to choose evaluation scenarios that align closely with the desired capabilities under assessment.

To this end, we provide a structured framework (see Table 2) that categorizes representative game environments based on their core interaction patterns and cognitive demands. These categories include basic social dilemmas, coordination and conflict resolution, competitive strategic reasoning, long-horizon planning, and social deduction and negotiation. Each game type highlights specific evaluation objectives, such as fairness, trust-building, opponent modeling, or multi-agent planning, thereby offering targeted benchmarks for assessing different dimensions of social competence. This categorization not only helps standardize evaluation protocols but also serves as a practical guide for selecting game scenarios tailored to particular research questions or development goals. By aligning game selection with evaluation objectives, researchers can more effectively assess the emergent behaviors, reasoning capabilities, and interactive robustness of social agents.

## 6 Future Directions

### 6.1 Standardized Benchmark Generation

The diversity and lack of standardization in current game types—often designed independently by developers with heterogeneous representations—pose significant challenges for the large-scale evaluation of social agents. This fragmentation makes it difficult to conduct efficient and reproducible benchmarking. Therefore, there is an urgent need for a standardized benchmark that offers broad coverage of game types, a consistent game description format, support for diverse agent architectures, and clearly defined evaluation metrics. Inspired

by platforms like OpenCompass (Contributors, 2023), such a benchmark should enable one-click evaluation by allowing users to configure the game environment, specify the agents to be tested, and select the desired evaluation metrics.

However, LLMs are typically pre-trained on vast amounts of data, which may include publicly available game datasets—raising concerns about data leakage and overfitting. To mitigate this issue, synthetic game data generation has emerged as a promising approach (Long et al., 2024). By leveraging classic game structures, LLMs can generate novel and diverse game scenarios through contextual reframing (Lorè & Heydari, 2024), producing out-of-distribution benchmarks that better evaluate an agent's generalization ability.

More concretely, two complementary strategies can be employed for scenario generation. From a structural perspective, developers can extract and manipulate the game's payoff matrix to construct new strategic settings while preserving core game mechanics. From a semantic perspective, LLMs can be used to reinterpret or re-describe existing games, generating alternative formulations that yield novel evaluation scenarios while maintaining logical coherence.

## 6.2   Reinforcement Learning Agents

Although current social agents have demonstrated promising performance across various game scenarios, existing research highlights notable limitations in multi-round, long-horizon, and complex multi-agent environments, where performance often degrades. This suggests that LLM-driven planning and decision-making alone are insufficient for achieving robust, scalable social intelligence. To address these challenges, future research should explore the integration of reinforcement learning (RL)—particularly multi-agent reinforcement learning (MARL)—to enhance state-space exploration, long-term adaptability, and emergent coordination.

MARL offers several insights that are highly relevant for improving LLM-based social agents. For instance, techniques such as centralized training with decentralized execution (CTDE) (Amato, 2024) can be used to guide LLM policy adaptation while preserving individual autonomy. Additionally, opponent modelling (He et al., 2016), credit assignment (Kazemnejad et al., 2024), and policy regularization (Cheng et al., 2019) in MARL can improve the agent's responsiveness to strategic variability and enhance generalization across diverse social contexts. However, integrating MARL into LLM training introduces new challenges. These include efficiency concerns, as LLMs are computationally intensive and may require specialized architectures or curriculum learning to reduce sample complexity; generalization gaps, especially when transferring learned behaviours across different social roles or task domains; and the need for consistent persona and belief modelling across episodes. Advancing this hybrid paradigm will also require fine-grained evaluation frameworks capable of tracing not just final performance but the underlying reasoning dynamics, theory-of-mind modelling, and role consistency throughout interactions.

## 6.3   Behaviour Pattern Mining

Existing studies primarily focus on predefined scenarios to examine the behaviour patterns of agents. However, with the advancement of multi-agent simulations, an intriguing direction is the automated discovery of game behaviour patterns that emerge spontaneously from agent interactions. It is important to note that, beyond explicit behaviours like cooperation, coordination, and betrayal, implicit causal relationships and long-term behavioural patterns should also be explored.

To mine such patterns, several methodological approaches can be leveraged. Unsupervised learning techniques, such as clustering and representation learning, can help identify latent behaviour categories and temporal motifs across trajectories (Rawassizadeh et al., 2016). Causal inference frameworks (e.g., Granger causality or structural causal models) can reveal inter-agent influence and dependency structures over time (Qiu et al., 2012). Additionally, trajectory segmentation and sequential pattern mining can be used to extract frequent decision sequences that correspond to strategic routines or social norms (Giannotti et al., 2007). Leveraging graph-based analysis of interaction networks can also shed light on evolving social roles and influence hierarchies within agent populations (Atzmueller, 2014). These approaches not only facilitate a deeper understanding of agents' behavioural preferences and latent traits but

also enable the study of how such patterns autonomously emerge—offering valuable insights for both AI and human behavioural research.

## 6.4 Pluralistic Game-Theoretic Scenarios

Although existing research has made notable strides across a wide range of game-theoretic scenarios, there remains a gap in the study of pluralistic game environments—settings that involve multiple languages, cultural norms, value systems, policies, and goals. These pluralistic scenarios introduce new layers of complexity, including behavioral preferences shaped by culturally grounded norms, value misalignment across agents, and belief conflicts arising from divergent objectives (Orner et al., 2024). Such dynamics pose unique challenges for the design and evaluation of socially intelligent agents and demand deeper exploration.

To develop robust pluralistic game-theoretic scenarios, several key desiderata should be considered: (1) Heterogeneity of agent profiles, including cultural, linguistic, and normative diversity; (2) Multi-objective frameworks, where agents pursue partially conflicting goals; and (3) Rich communicative channels, enabling nuanced language use, code-switching, or culturally specific cues. Evaluating agents in these settings requires multi-faceted metrics. In addition to task performance, evaluations should account for norm sensitivity, value alignment, cross-cultural adaptability, and the agent's ability to mediate or negotiate among conflicting belief systems. Metrics such as cultural appropriateness, interaction fluency, and conflict resolution success can serve as important complementary indicators. Scenario generation can be approached in two ways: from a knowledge-based perspective, designers can draw from real-world policy conflicts, international relations, or sociocultural theory to construct grounded simulation environments. From a data-driven perspective, large language models can be used to simulate role-play dialogues or generate scenarios by conditioning on demographic or cultural descriptors, yielding diverse and customizable pluralistic environments.

# 7 Related Works

The human-like capabilities of LLMs have drawn significant attention from social science researchers, prompting extensive exploration at the intersection of AI and social sciences (Xu et al., 2024a). A key development in this area is the shift from traditional Agent-Based Modeling to LLM-based agents, as explained by Ma et al. (2024) through computational experiments. Numerous studies have since applied LLM-based agents to diverse game scenarios, such as poker, Minecraft, and DOTA II, with more detailed summaries provided by (Xu et al., 2024b; Hu et al., 2024b;a). Furthermore, Zhang et al. (2024c) have analyzed the core strategic reasoning capabilities of these agents, distinguishing them from other reasoning approaches. While the previous reviews provide comprehensive overviews of related fields, our survey specifically focuses on social agents equipped with beliefs, preferences, and reasoning capabilities within diverse game-theoretic scenarios.

# 8 Conclusion

We provide a comprehensive summary of existing research on LLM-based social agents in game-theoretic scenarios from three perspectives: game framework, social agents, and evaluation protocol. This interdisciplinary field covers a wide range of topics, including social sciences, economics, decision sciences, and theory of mind. Current studies have primarily explored the more direct external behavioural patterns and internal cognition of social agents. Therefore, future research should focus on developing theoretical frameworks for cognitive representations within LLMs, conducting in-depth analyses of implicit and long-term game behaviour patterns, and enhancing agents' reasoning and planning capabilities in dynamic environments.

**Broader Impact Statement**

Developing agents with advanced social intelligence is one of the ultimate goals of artificial intelligence. On one hand, such agents demonstrate enhanced collaboration, a deeper understanding of mental states, and seamless integration into human society. On the other hand, negative social behaviors may also emerge, such as deception, malicious competition, and verbal aggression, which conflict with the vision of a harmonious human-AI coexistence.

Therefore, we carefully examine the potential negative impacts that social agents may have on human society, serving as a cautionary perspective for future social agent development. One major concern is *deception and manipulation*, where agents may bluff or mislead to achieve strategic goals. They may also engage in *malicious competition*, exploiting others to gain advantage, or exhibit *verbal and social aggression*, such as generating insults or polarizing language. Additionally, social agents can *amplify societal biases*, leading to discriminatory behaviors, and contribute to the *erosion of trust*, especially when users struggle to distinguish genuine human interactions from artificial ones. These agents may further *undermine human autonomy* by subtly steering decisions through persuasion, often without transparency. Due to their *scalability of harm*, even a single flawed agent can rapidly propagate misinformation or harmful behaviors across platforms. Moreover, the risk of *impersonation and infiltration* arises when agents mimic human users, potentially deceiving communities or individuals. These challenges highlight the critical need for careful design, value alignment, and robust supervision in the development and deployment of socially intelligent agents.

We now categorize the development and deployment of social agents into four stages:(1) Designing social agents, (2) Evaluating social agents, (3) Deploying social agents, and (4) Supervising social agents. Accordingly, we discuss the potential risks and feasible mitigation strategies for each stage. *Design Phase*: The underlying algorithms determine the agent's behavioral preferences. Poorly designed algorithms may inadvertently lead to negative behaviors. To address this, researchers should enhance alignment algorithms, including safety alignment and moral alignment, to mitigate these risks at a fundamental level. Another promising approach is the design of behavioral plugins, where small models trained as plug-and-play behavior controllers can regulate agent actions dynamically. *Evaluation Phase*: Rigorous evaluation is crucial before deploying social agents in real-world applications. Agents exhibiting negative behaviors should be prevented from entering the deployment phase. One effective approach is to evaluate social agents across diverse game scenarios, allowing for a benchmarking framework that assesses their behavioral preferences under dynamic conditions. *Deployment Phase*: Direct large-scale deployment may lead to unforeseen negative consequences that were not observed in smaller-scale testing. Therefore, social agents should first be deployed in low-risk, small-scale environments, with a gradual expansion in scope and scale to monitor anomalies in real time. *Supervision Phase*: Effective oversight of social agents is essential. This can be achieved by designing automated monitoring systems that enable large-scale real-time surveillance. Behavioral analysis can be used to issue early warnings, assisting human supervisors in decision-making.

Additionally, it is important to note that most of the studies referenced in this paper utilize the GPT series as the large language model, which limits the generalizability of the experimental results. Differences in model architectures, training data, and alignment techniques can significantly impact the behavioral patterns exhibited by different models. Future research should explore a broader range of large language models, such as Claude, Gemini, Llama, and DeepSeek, to derive more comprehensive and reliable conclusions.

### Acknowledgments

This work was supported by Hong Kong Innovation and Technology Support Programme Platform Research Project fund (ITS/269/22FP).

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
