# OpenReview forum: "A Survey on Large Language Model-Based Social Agents in Game-Theoretic Scenarios"
_TMLR — Accepted by TMLR_

### Review · Reviewer_uaWb · 2025-02-01

**Summary Of Contributions:**

This survey examines LLM-based social agents in game-theoretic scenarios, covering three key aspects: Game Framework, Social Agents, and Evaluation Protocol. For each aspect, it provides a comprehensive review of related work, organizing it in a problem-oriented structure. Additionally, it presents the authors' perspectives on current challenges and limitations in the field.

**Audience:**

Yes

**Claims And Evidence:**

Yes

**Requested Changes:**

1. In line 6, paragraph 4 of Section 1, "*... such as classic game-theoretic and poker ...*". I think replacing "classic game-theoretic" with a more concrete example, such as "prisoner's delimma" would be better. "classic game-theoretic" is not straighforward to understand or imagine.
2. In Section 3, it would be helpful to include a description or a pipeline illustrating how the three modules (Preference, Belief, and Reasoning) interact during agent activity.
3. In Figure 7, the two reasoning methods are presented side by side, which may lead readers to unconsciously compare them. To facilitate a more meaningful comparison, it would be better if both methods were applied to the same game scenario, making their tactical differences more apparent. If they are not directly comparable, the authors could clarify this in the figure caption, noting that "Theory of Mind reasoning" is particularly suited for multi-agent scenarios, while "RL reasoning" is used for dynamic environments.
4. The authors might also consider adding a discussion in Section 4 on agent performance across different game scenarios.

**Strengths And Weaknesses:**

**Strengths**:
This survey is well-written and clear. It provides a comprehensive review of relevant work in the field, with detailed descriptions of each study. I also appreciate the "Takeaways" sections at the end of each part, which help readers reflect on key research problems in a timely manner.

**Weaknesses**:
I don’t see major issues with this survey. However, one weakness is that the three modules discussed in the "Social Agent" section—Preference, Belief, and Reasoning—feel somewhat scattered and disconnected. It would be helpful to include a discussion on how these modules interact during agent activity or their roles in the decision-making process. In my opinion, the lack of connections between them is noticeable, as agents' cognition and actions rely on an interplay of these components. For instance, can the Preference and Belief modules influence Reasoning, and vice versa? Clarifying these relationships would strengthen the coherence of this section.

Another weakness is the lack of an overview of the current performance of game agents. While I understand that providing quantitative results or direct performance comparisons is challenging, some qualitative insights would be valuable. For example, how does agent performance compare to human players? What game scenarios pose the greatest challenges for agents? Addressing these questions would give readers a better understanding of the progress of existing game agents research.

---

> ### Author Response · Authors · 2025-03-22
> **Responses to Reviewer uaWb**
>
> Thank you for your prompt and insightful feedback. Below, we provide detailed point-by-point responses to your comments. All corresponding revisions have been highlighted in yellow in the latest version of the manuscript.
>
> ---
>
> **Lack of description of connections among the three modules (preference, belief, and reasoning).**
>
> **Answer:** Thank you for your constructive feedback. We fully agree that the interactions among the three modules play a crucial role in the final decision-making process. To address this, we have added Section 3.4: *PBR-Triangular Interaction*, which specifically discusses the interaction between Preference, Belief, and Reasoning.
>
> In summary, we first define the interaction dynamics among the three modules and their mutual influences. Then, we provide a detailed explanation of six types of interactions, elaborating on their definitions and impact on decision-making. To enhance clarity, we further illustrate these interactions with concrete examples. Finally, in the Takeaways, we discuss the importance of designing context-adaptive flows and automated scheduling algorithms for module interactions.
>
> **Revisions in the Manuscript:**
>
> 1. **Added Section 3.4**: PBR-Triangular Interaction (`Lines 350–394`)
> 2. **Introduced Figure 9**, illustrating the interaction diagram of the three modules (`Figure 9`).
>
> ---
>
> **Lack of an overview of the current performance of game agents.**
>
> **Answer:** We appreciate this valuable suggestion. The lack of performance discussion indeed makes it difficult for readers to capture the current progress in social agent research. To address this issue, we have added Section 4.3: *Performance Assessment of Social Agents*, where we analyze the performance of social agents in various game scenarios.
>
> As you correctly pointed out, the diversity of game environments and evaluation metrics makes comprehensive quantitative analysis challenging. To mitigate this, we introduce a new metric, *Score Rate*, which represents an agent’s performance relative to the highest achievable performance in a given game. Based on an extensive review of existing studies, we analyze the strengths and weaknesses of social agents in choice-focusing and communication-focusing games, compare their performance across different games, and evaluate their capabilities relative to human players. Finally, in the Takeaways, we discuss the importance of designing standardized evaluation metrics and incorporating human players in performance comparisons.
>
> **Revisions in the Manuscript:**
>
> 1. **Added Section 4.3**: Performance Assessment of Social Agents (`Lines 439–466`).
> 2. **Introduced Table 1**, summarizing the performance of social agents across different games (`Table 1`).
>
> ---
>
> **Lack of clarity in the comparison of the two reasoning methods in Figure 7.**
>
> **Answer:** Thank you for your valuable suggestion. The two reasoning methods—Theory-of-Mind reasoning and Reinforcement Learning-style reasoning—are independent approaches; however, they can be integrated to enhance reasoning effectiveness in complex game-theoretic scenarios.
>
> To address the clarity issue in Figure 7, we have revised the figure and introduced Hybrid-form Reasoning, explicitly illustrating how these two reasoning methods can be combined for improved strategic decision-making.
>
> **Revision in the Manuscript:**
>
> 1. **Updated Figure 8 (previous fig 7)**, depict the two commonly used reasoning methods in strategic reasoning, along with a hybrid reasoning approach that integrates both (`Figure 8`).
>
> ---
>
> **Writing revision suggestions.**
>
> **Answer:** We agree with your suggestion that replacing “classic game-theoretic” with “Prisoner’s Dilemma” allows readers to more intuitively and accurately understand the type of game being described.
>
> **Revision in the Manuscript:**
>
> 1. Replaced classic game-theoretic with Prisoner’s Dilemma (`Line 52`).
>
> ---
>
> We appreciate your insightful feedback, which has helped enhance the clarity and completeness of our survey.

---

> > ### Comment · Reviewer_uaWb · 2025-03-22
> > **Thank authors for their response**
> >
> > I reviewed the updated version and appreciate the addition of subsections 3.4 and 4.3, as well as the revision of Figure 7. These updates address my earlier concerns and integrate well with the overall context.
> >
> > However, I have a couple of further suggestions regarding section 4.3. First, the authors state that "a score of 60 is widely recognized as the passing threshold" for the "Score Rate" metric. It would be helpful to include references or citations to support this claim. Second, I recommend renaming "Score Rate" to a more intuitive term, such as "Relative Agent Score."

---

> > > ### Author Response · Authors · 2025-03-24
> > > **Responses to Reviewer uaWb**
> > >
> > > Thank you for your prompt response and further suggestions.
> > >
> > > Regarding the first point, we have added a citation to *Performance of ChatGPT on USMLE: Potential for AI-assisted medical education using large language models* (Kung et al., 2023), a widely cited paper (over 3,000 citations) which identifies 60% accuracy as the passing threshold for the USMLE. This serves as a reference to support our evaluation threshold.
> > >
> > > Regarding the second point, we agree that *Relative Agent Score* is a more appropriate term. Given that the preceding column in Table 1 already reports the *Agent Score*, the use of *Relative Agent Score* better emphasizes the comparison. We have revised the manuscript accordingly.
> > >
> > > **Revisions in the Manuscript:**
> > >
> > > 1. Added citation to *(Kung et al., 2023)* (`Line 450`).
> > > 2. Replaced *Score Rate* with *Relative Agent Score* throughout the text and Table 1 (`Lines 446, 449 and Table 1 Caption`).

---

### Review · Reviewer_dmrB · 2025-03-09

**Summary Of Contributions:**

# Summary
This paper is a survey on multi-llm-agent literature with a specific focus on game theory inspired settings [^1].

It discusses the current work in this area through three lens:

1. Game Framework: choice (non-verbal) focused, or communication (verbal) focused
2. Agent Design: preference, belief, and reasoning modules
3. Evaluation: outcome evaluation, and behavior evaluation

## Some key takeaways

### Game Framework
1. Current research lacks unified evaluation frameworks
2. Data contamination is a concern as classic games may be in pre-training data
3. Limited exploration of "strategic behavior" through "action language" in games with minimal verbal communication
4. More realistic and diverse games promote greater agent behavior diversity
5. Need for better process evaluation mechanisms beyond just outcomes

### Social Agent
1. Preference: LLMs show some ability to follow preferences but struggle with less common preferences
2. Belief: Ongoing debate about whether LLMs possess beliefs; models show limited and unstable belief revision capabilities
3. Reasoning: Theory-of-Mind reasoning in LLMs remains contentious; Reinforcement Learning combined with search techniques shows promise

### Evaluation Protocol
1. Win rate alone is insufficient; researchers propose expanded metrics like efficiency-adjusted win rate and comeback win rate
2. Need for a comprehensive evaluation metrics system that is hierarchical, abstract, and quantifiable

## Contributions

1. A good survey for people who are getting into this field to understand the latest research.
2. Pointing out important future directions including pluralistic scenarios with diverse goals, cultures, and values

[^1]: Although the definition of game theoretical scenarios is not clearly defined, e.g. role-playing goal-oriented scenarios like Sotopia is not included in this study, but I guess the authors are focusing on the cases where the rules for win and lose are clearly defined.

**Audience:**

Yes

**Broader Impact Concerns:**

Agents that learn to deceive or compete with humans should be studied with caution. So a Broader Impact Statement should be provided to discuss potential harms when the agents become socially intelligent.

**Claims And Evidence:**

Yes

**Requested Changes:**

Please expand Section 5 to provide sketches with references regarding the potential future directions:

1. Standardized Benchmark Generation: What are the desiderata for a standardized benchmark? How should we evaluate agents in this benchmark? How to generate the scenarios?
2. Reinforcement Learning Agents: What insights in MARL should be integrated into LLM agent training? What are the challenges? Efficiency? Generalization?
3. Behaviour Pattern Mining: What are the potential methods to mine these behavior patterns?
4. Pluralistic Game-Theoretic Scenarios: Similar to 1.

This is still a new field, so I appreciate the authors' effort to put together a survey to try to unify the literature. However, a survey should go beyond summarization, but also provides new insights that one could not easily get by reading the summaries.

**Strengths And Weaknesses:**

## Strengths

1. This paper is well-structured with the three lens mentioned above covering the important aspect of the research scope as well as boxes takeaways with discussion.
2. The papers covered in the survey are recent and timely.


## Weaknesses

1. It is hard to evaluate a survey's contribution to the knowledge of practitioners. After reading it, I don't think this paper provides any novel views after summarizing the papers. The completeness of the survey makes it especially hard for readers to grasp key insights.
2. What are the fundamental problems of LLMs when they are applied to these game-theoretical scenarios? This paper provides some "balanced" views, e.g. LLMs could have belief or not have belief, without strong arguments for either.
3. I think Section 5, is more important than other sections, since it points out potential directions in this field. It should be expanded.

---

> ### Author Response · Authors · 2025-03-22
> **Responses to Reviewer dmrB**
>
> Thank you for your insightful and constructive feedback. Below, we provide detailed point-by-point responses to your comments. All corresponding revisions have been highlighted in yellow in the latest version of the manuscript.
>
> ---
>
> **Lack of actionable guidance.**
>
> **Answer:** We sincerely thank you for identifying this important issue. In the original submission, we indeed overlooked the need to distill existing knowledge into concrete guidance for future development and evaluation of social agents. To address this, we have added a new section 5 titled *Practical Guides for Researching Social Agents*, which synthesizes insights from the literature and offers actionable recommendations for both the design and evaluation of social agents. Specifically, for design, we highlight the importance of the Preference, Belief, and Reasoning modules, along with their interactions, and further elaborate on the practical challenges faced in implementing each component. For evaluation, we provide a structured table outlining key game categories, core evaluation focuses, and the specific capabilities being assessed, thereby offering a practical reference for researchers to select suitable game scenarios based on their evaluation goals.
>
> **Revisions in the Manuscript:**
>
> 1. **Added Section 5**: offers concrete guidance on both the design and evaluation of social agents  (`Lines 467–524`).
> 2. **Introduced Table 2**: guidelines for selecting game scenarios in social agent evaluation (`Table 2`)
>
> ---
>
> **The “Future Work” section needs further development.**
>
> **Answer:** Thank you for your thoughtful suggestion. In response to your comments, we have revised the *Future Directions* section to address each point in detail. Specifically, we expanded the discussion of key research challenges and proposed feasible implementation strategies, aiming to help readers more easily identify promising future directions and select actionable approaches for further exploration.
>
> **Revision in the Manuscript:**
>
> 1. **Revised Section 6***:* elaborate on specific research aspects and present concrete, implementable suggestions (`Lines 525–599`).
>
> ---
>
> **Lack of broader impact section.**
>
> **Answer:** Thank you for pointing out this important omission. We fully agree that discussing the broader impact of social agents is essential. In response, we have added a dedicated section titled *Broader Impact Statement.* In this section, we first outline the potential risks posed by social agents, including Deception & Manipulation, Malicious Competition, Verbal Aggression, Bias & Discrimination, Erosion of Trust, Undermining Autonomy, Scalability of Harm, and Impersonation & Infiltration. We then examine these risks and corresponding mitigation strategies across the four key stages of social agent development and deployment:
>
> - Design Phase: Developing robust alignment algorithms.
> - Evaluation Phase: Assessing agent behavior in diverse game-theoretic scenarios.
> - Deployment Phase: Starting with small-scale, low-risk environments before broader rollout.
> - Supervision Phase: Implementing automated monitoring systems for real-time behavior analysis and intervention.
>
> Finally, we emphasize that most current studies rely heavily on GPT-series models, which may limit the generalizability of findings. To address this, we encourage the research community to explore a broader range of foundation models, such as Claude, Gemini, Llama, and DeepSeek, to obtain more comprehensive and reliable insights.
>
> **Revision in the Manuscript:**
>
> 1. **Added**  **Broader Impact Statement**, discussing the potential risks of social agents and corresponding mitigation strategies (`Lines 618–657`).
>
> ---
>
> **The insight of this survey should be enhanced.**
>
> **Answer:** We fully agree that a survey should go beyond mere summarization and enumeration—it should also provide meaningful insights to guide future research. In the revised version, we have strengthened the survey in four key aspects compared to the original submission:
>
> 1. A well-structured literature taxonomy to support future research positioning;
> 2. A unified and comprehensive performance comparison to highlight the strengths and weaknesses of current social agents (`added Section 4.3: Performance Assessment of Social Agents`);
> 3. Detailed development guidelines to inform future agent design and evaluation (`added Section 5: Practical Guides for Researching Social Agents`);
> 4. Concrete future directions to outline promising research avenues (`expanded Section 6: Future Directions`).
>
> Through these enhancements, we aim to provide not only a structured summary of the field but also actionable insights for researchers.
>
> **Revision in the Manuscript:**
>
> 1. **Added a contribution summary** in the final paragraph of the *Introduction* to highlight the core takeaways of this survey and help readers quickly identify the most relevant content (`Lines 72–80`).

---

> > ### Author Response · Authors · 2025-03-22
> > **Responses to Reviewer dmrB**
> >
> > **Additional references.**
> >
> > **Answer:** To ensure the timeliness of our survey, we have incorporated two additional papers that are highly relevant to our study:
> >
> > 1. PokerBench: Training Large Language Models to Become Professional Poker Players (`Lines 122–125`).
> > 2. Barriers and Pathways to Human-AI Alignment: A Game-Theoretic Approach (`Line 267`).
> >
> > ---
> >
> > We appreciate your insightful feedback, which has helped enhance the clarity and completeness of our survey.

---

> ### Author Response · Authors · 2025-03-25
> **Looking Forward to Further Discussion on the Revision**
>
> We sincerely appreciate your valuable feedback and have revised the manuscript accordingly. We look forward to any further comments you may have and would be glad to continue the discussion to further refine the paper within the review cycle.

---

### Review · Reviewer_xzPJ · 2025-03-16

**Summary Of Contributions:**

This paper is survey about evaluating the social intelligence of Language Agents by making LLMs play social games. The litterature review is split in three parts: 1) what are the existing game frameworks, 2) what are the common architecture designs for such social agents, and 3) what are the main evaluation criteria used to evaluate social intelligence. The authors provide a taxonomy of the field mapping recent works to different research topics. They also make use of several figures to support and explain the different topics discussed in this paper. Overall, this survey paper aims at depicting the recent research in this field as accurately as possible and provides many takeways one can act upon to extend it.

As a researcher in an adjacent field, it is my impression that I got a great deal of information from this survey paper. I'm certain other in the TMLR community will get something out of reading it as well. I recommend this paper for acceptance.

**Audience:**

Yes

**Broader Impact Concerns:**

I encourage the authors to include a Broader Impact section. For instance, what are the implications of developing such Social Agents and making them play against/with humans? Are those game-theoretic frameworks a valid approach to measure LLMs ability to deceive people?

Also, it is my understanding that most the recent research papers covered in this survey focus on evaluation of the GPT-family models. While OpenAI's models were the first to be "easily" accessible to the research community, I'm sure recent works must have considered other models as well, e.g. Claude, Gemini, and even the open-weights ones like Llama? If that's not the case, I believe this bias should be pointed out in this survey. In my opinion, this is a valid takeaway and would encourage the community to diversify their model choices.

**Claims And Evidence:**

Yes

**Requested Changes:**

### Typos
- Figure 3 (left). "the payoff matric" -> "... metric"
- Figure 3 (middle). "your best size" -> "... bet size"

**Strengths And Weaknesses:**

### What I like about this paper

- The paper is well written and well-structured paper. The authors go beyond simply listing the recent works, each section ends on a Takeaways section which is exactly what I look for when reading survey papers.
- Figure 2 provide a nice taxonomy of many of the references used for this survey paper.
- I appreciated the many figures illustrating concrete examples of game scenarios and social agent modules. That helped me understand
- As someone outside the game-theoretic social scenarios field, I feel I have acquired valuable insights and knowledge on that topic. To me, this survey is a good starting point to start digging deeper.

### Opportunities for improvement
- I feel the survey could do a better job a unifying the different terms used in different papers and that seem to all point to the same thing. For instance, "AI personas" (Section 2.1), "social behaviours" (Section 2.2), "traits", "preferences" (Section 3.1), etc. If they are not the same, I believe a comparison between those would be appreciated by the readers.

- When writing papers, I found there is a delicate balance between over-providing term definitions vs. referring the readers to external sources. I do believe survey papers (especially journal papers) should aim to be self-contained as much as possible by at least providing a one sentence summary definition and provide reference for more in-depth details. For instance, I had to look up the Matthew Effect. Another example, when mentioning "prisoner's dilemma" is simply refered as "the most famous and recognized game". A brief explanation would go a long way, or in this specific case, the authors could refer the readers to Fig 3 since there's actually an example.

---

> ### Author Response · Authors · 2025-03-22
> **Responses to Reviewer xzPJ**
>
> Thank you for your appreciation and valuable suggestions. Below, we provide detailed point-by-point responses to your comments. All corresponding revisions have been highlighted in yellow in the latest version of the manuscript.
>
> ---
>
> **Lack of broader impact section.**
>
> **Answer:** Thank you for pointing out this important omission. We fully agree that discussing the broader impact of social agents is essential. In response, we have added a dedicated section titled “Broader Impact Statement.”
>
> In this section, we first outline the potential risks posed by social agents, including Deception & Manipulation, Malicious Competition, Verbal Aggression, Bias & Discrimination, Erosion of Trust, Undermining Autonomy, Scalability of Harm, and Impersonation & Infiltration. We then examine these risks and corresponding mitigation strategies across the four key stages of social agent development and deployment:
>
> - Design Phase: Developing robust alignment algorithms.
> - Evaluation Phase: Assessing agent behavior in diverse game-theoretic scenarios.
> - Deployment Phase: Starting with small-scale, low-risk environments before broader rollout.
> - Supervision Phase: Implementing automated monitoring systems for real-time behavior analysis and intervention.
>
> Finally, we emphasize that most current studies rely heavily on GPT-series models, which may limit the generalizability of findings. To address this, we encourage the research community to explore a broader range of foundation models, such as Claude, Gemini, Llama, and DeepSeek, to obtain more comprehensive and reliable insights.
>
> **Revisions in the Manuscript:**
>
> 1. **Added**  **Broader Impact Statement**, discussing the potential risks of social agents and corresponding mitigation strategies (`Lines 618–657`).
>
> ---
>
> **The descriptions in the paper need to be self-contained.**
>
> **Answer:** We fully agree with your perspective that a survey paper should be self-contained, ensuring clarity and accessibility for readers. To enhance the self-contained nature of our manuscript, we have implemented the following improvements: We have provided descriptions and simple examples for the key game types discussed in the paper, including Classic game-theoretic games, Poker, and Auction, allowing readers to more intuitively understand the game scenarios under discussion. Additionally, we have elaborated on the Matthew Effect to enhance clarity and ease of understanding for readers unfamiliar with the concept.
>
> **Revisions in the Manuscript:**
>
> 1. **Added Figure 4**, introducing different types of game-theoretic games. (`Figure 4`).
> 2. Expanded and refined the description of the **Matthew Effect** for improved readability (`Lines 176-178`).
>
> ---
>
> **Terms with the same meaning need to be standardized.**
>
> **Answer:** Thank you for your careful suggestion. We conducted a thorough review of the manuscript and found that personas, traits, and preferences were used interchangeably to refer to the same concept—preferences. To ensure consistency, we have standardized the terminology by using preference throughout the paper. Additionally, since social behaviors specifically refer to observable actions and interactions, we have included a corresponding clarification in the manuscript to enhance the clarity of its meaning.
>
> **Revisions in the Manuscript:**
>
> 1. **Standardized terminology**: Replaced persona and trait with preference throughout the text  (`Lines 133, 135, 241, 265`).
> 2. **Added clarification** to improve the definition and understanding of social behaviors  (`Lines 171-173`).
>
> ---
>
> **Writing revision suggestions.**
>
> **Answer:** Thank you for your thorough review. We have revised Figure 3 accordingly.
>
> **Revisions in the Manuscript:**
>
> 1. **Updated Figure 3**, correcting spelling errors (`Figure 3`).
>
> ---
>
> We appreciate your insightful feedback, which has helped enhance the clarity and completeness of our survey.

---

> > ### Comment · Reviewer_xzPJ · 2025-03-24
> > **Great additions**
> >
> > Thank you Authors for all the changes: adding Broader Impact Statement section (amongst others), adding clarifying figure and definitions, and uniformizing the terms. I appreciated the highlighting to see the changes in the new revision.
> >
> > To me, the paper has been strengthened and recommend for acceptation.

---

> > > ### Author Response · Authors · 2025-03-24
> > > **Responses to Reviewer xzPJ**
> > >
> > > Thank you for your kind feedback and recommendation. We’re glad the revisions helped clarify and strengthen the paper!

---

### Decision · Action_Editor_ccSs · 2025-04-20

**Recommendation:** Accept as is

**Comment:**

The authors have expanded their work to a broader literature, recent publications, and included guidance for future work in the area.

**Audience:**

The social agents literature and more generally the use of LMs for collaboration, game playing, negotiation, and more.

**Claims And Evidence:**

This work outlines a taxonomy of the social agent literature, organizing recent papers to identify challenges and limitations. The framework and examples are well covered and specific domains/categories are also illustrated. The reviewers have all indicated that the current manuscript is updated to address their concerns.